# Characterizing the dynamics underlying global spread of epidemics

Lin Wang[1] & Joseph T. Wu [1]

Over the past few decades, global metapopulation epidemic simulations built with worldwide air-transportation data have been the main tool for studying how epidemics spread from the origin to other parts of the world (e.g., for pandemic influenza, SARS, and Ebola). However, it remains unclear how disease epidemiology and the air-transportation network structure determine epidemic arrivals for different populations around the globe. Here, we fill this knowledge gap by developing and validating an analytical framework that requires only basic analytics from stochastic processes. We apply this framework retrospectively to the 2009 influenza pandemic and 2014 Ebola epidemic to show that key epidemic parameters could be robustly estimated in real-time from public data on local and global spread at very low computational cost. Our framework not only elucidates the dynamics underlying global spread of epidemics but also advances our capability in nowcasting and forecasting epidemics.

---

[1] WHO Collaborating Centre for Infectious Disease Epidemiology and Control, School of Public Health, Li Ka Shing Faculty of Medicine, The University of Hong Kong, 7 Sassoon Road, Hong Kong Special Administrative Region, 999077 Pokfulam, China. Lin Wang and Joseph T. Wu contributed equally to this work. Correspondence and requests for materials should be addressed to J.T.W. (email: joewu@hku.hk)

Since the 1980s, metapopulation epidemic models built with worldwide air-transportation network (WAN) data have been the main tool for studying global spread of epidemics, such as pandemic influenza[1-4], SARS[5,6], MERS-CoV[7], Ebola[8], and Zika[9,10]. The complexity of these models has substantially grown over the past few decades, advancing from 55 populations in the Rvachev–Longini model in 1985[1] to more than 3500 populations in the state-of-the-art simulator GLEAM powered by supercomputer[11,12]. Despite the long history and widespread use of these models[13-17], most studies on global spread of epidemics have relied on computationally intensive simulations that provide limited epidemiologic insights, whereas an analytical understanding of the underlying epidemic dynamics has only been partially elucidated in recent years[18-20]. Here, we build on these recent advancements and develop a novel framework for analytically characterizing how epidemic arrivals for different populations around the world depend on the epidemiologic parameters and structure of the WAN. We first validate this framework using global epidemic simulations. We then illustrate its potential to enhance our ability to nowcast and forecast epidemics by applying it retrospectively to the 2009 influenza A/H1N1 pandemic and the 2014 West African Ebola epidemic in Liberia.

## Results

**Major assumptions in the framework.** Throughout this paper, we consider only global spread of epidemics with relatively fast timescales in which epidemics in each population peak within 300 days after establishment (e.g., pandemic influenza, MERS, Ebola) such that changes in demographics (e.g., births, aging) is negligible. In metapopulation epidemic models, populations (e.g., cities) around the world are connected through the travel of individuals via the WAN (see WAN metapopulation epidemic model in Methods for details). We designate population $i$ as the epidemic origin which is seeded with $s_i$ infections at time 0. For any given population $j$, we denote its population size by $N_j$ and initial epidemic growth rate by $\lambda_j$. If populations $j$ and $k$ are directly connected in the WAN, the mobility rate from population $j$ to $k$ is defined as $w_{jk} = F_{jk}/N_j$, where $F_{jk}$ is the direct air-traffic (passengers per day) and $w_{jk}$ ranges mostly between $10^{-6}$ and $10^{-3}$ per day in the WAN (Supplementary Fig. 1). Let $T_{ij}^n$ be the time at which population $j$ receives its $n$th imported infection. The epidemic arrival time (EAT) for population $j$ is defined as $T_{ij}^1$. Our framework is built upon the following assumption[19,21].

Assumption 1: Suppose populations $j$ and $k$ are directly connected in the WAN and only population $j$ is infected. Exportation of infections from population $j$ to $k$ is a non-homogeneous Poisson process (NPP)[22] with intensity function $w_{jk}I_j(t)$ where $I_j(t)$ is the disease prevalence (number of infectives) in population $j$ at time $t$ (see Details on assumption 1 in Methods for details).

Supplementary Figure 2 shows that assumption 1 is very accurate for a wide range of plausible epidemic scenarios. We will show that the dynamics of global spread is largely analytically tractable because the following assumption is also accurate across these same scenarios.

Assumption 2: After the epidemic has established in a given population $j$, the first few exportations occur while disease prevalence is still growing exponentially, i.e., $I_j(t) = s_j \exp(\lambda_j t)$.

To this end, we progressively build up our framework by characterizing the probability distribution of EATs for all populations in three metapopulation networks with increasingly complex structure: (i) The two-population network which has the simplest metapopulation structure; (ii) the shortest-path-tree of the WAN (WAN-SPT hereafter) which is the dominant subnetwork driving global spread of epidemics as described by the seminal study by Brockmann and Helbing[20]; and (iii) the WAN.

**The two-population network.** In the two-population network, the origin population $i$ is only connected to population $j$. Under assumption 2, the probability density function (pdf) of $T_{ij}^n$ can be expressed in closed-form:

$$f_n(t|\lambda_i, \alpha_{ij}) = \left(\frac{\exp(\lambda_i t) - 1}{\lambda_i}\right)^{n-1} \frac{\alpha_{ij}^n}{(n-1)!} \exp\left[\lambda_i t - \frac{\alpha_{ij}}{\lambda_i}(\exp(\lambda_i t) - 1)\right], \quad (1)$$

where $\alpha_{ij} = w_{ij}s_i$, which we term adjusted mobility rate. Figure 1 shows that if $n$ is smaller than 10, Eq. 1 is accurate across a wide range of realistic scenarios (e.g., the percent error in expected EAT is uniformly below 2%), which correspond to epidemics ranging from pandemic influenza (with doubling time around 4 −5 days) to Ebola (with doubling time longer than 20 days). This result leads to the following corollaries for the WAN-SPT and WAN analysis: (i) Exportation of the first $n$ infections is essentially an NPP with intensity function $\alpha_{ij} \exp(\lambda_i t)$; and (ii) the expected time of the $n$th exportation is given by $E\left[T_{ij}^n\right] = \frac{1}{\lambda_i}\exp\left(\frac{\alpha_{ij}}{\lambda_i}\right)\sum_{m=1}^{n} E_m\left(\frac{\alpha_{ij}}{\lambda_i}\right)$, where $E_m(x)$ is the exponential integral. Hence,

$$E\left[T_{ij}^n \Big| T_{ij}^1\right] = T_{ij}^1 + \frac{1}{\lambda_i}\exp\left(\frac{\alpha_{ij}\exp\left(\lambda_i T_{ij}^1\right)}{\lambda_i}\right) \sum_{m=1}^{n-1} E_m\left(\frac{\alpha_{ij}\exp\left(\lambda_i T_{ij}^1\right)}{\lambda_i}\right), \quad (2)$$

which corresponds to the $(n-1)$th exportation for an epidemic that starts at time $T_{ij}^1$ with seed size $s_i \exp\left(\lambda_i T_{ij}^1\right)$.

These analytics can be used to formulate closed-form likelihood functions for inferring parameters from disease surveillance and global spread data (see Methods).

**WAN-SPT.** For the WAN-SPT and WAN analysis, we use the worldwide passenger booking data from the Official Airline Guide (OAG) and the Gridded Population of the World data set (Version 4) from the NASA Socioeconomic Data and Applications Center at Columbia University to build a stochastic metapopulation global epidemic simulator with 2309 populations and 54,106 connections (see The global epidemic simulator in Methods). This simulator is similar to GLEAM but without the effect of local commuting which has negligible impact on global spread[23]. Brockmann and Helbing[20] suggested that global spread of epidemics is primarily driven by the WAN-SPT subnetwork in which each population is connected to the epidemic origin via only one path. We will show that for each population $k$ in the WAN-SPT, the time at which the $n$th importation occurs, namely $T_{ik}^n$, can be well characterized by $f_n(t|\lambda, \alpha)$ (Eq. 1), where $\lambda$ and $\alpha$ are specifically parameterized to account for the hub-effect and continuous seeding (explained in the next two sections and Fig. 2a, b). This provides a profound insight: the epidemic arrival process for each population $k$ in the WAN-SPT can be approximated as an NPP with intensity function in the form of $\alpha \exp(\lambda t)$.

*Hub-effect*: Hubs are populations that have direct connections to many populations in the WAN, e.g., Hong Kong, Beijing, New

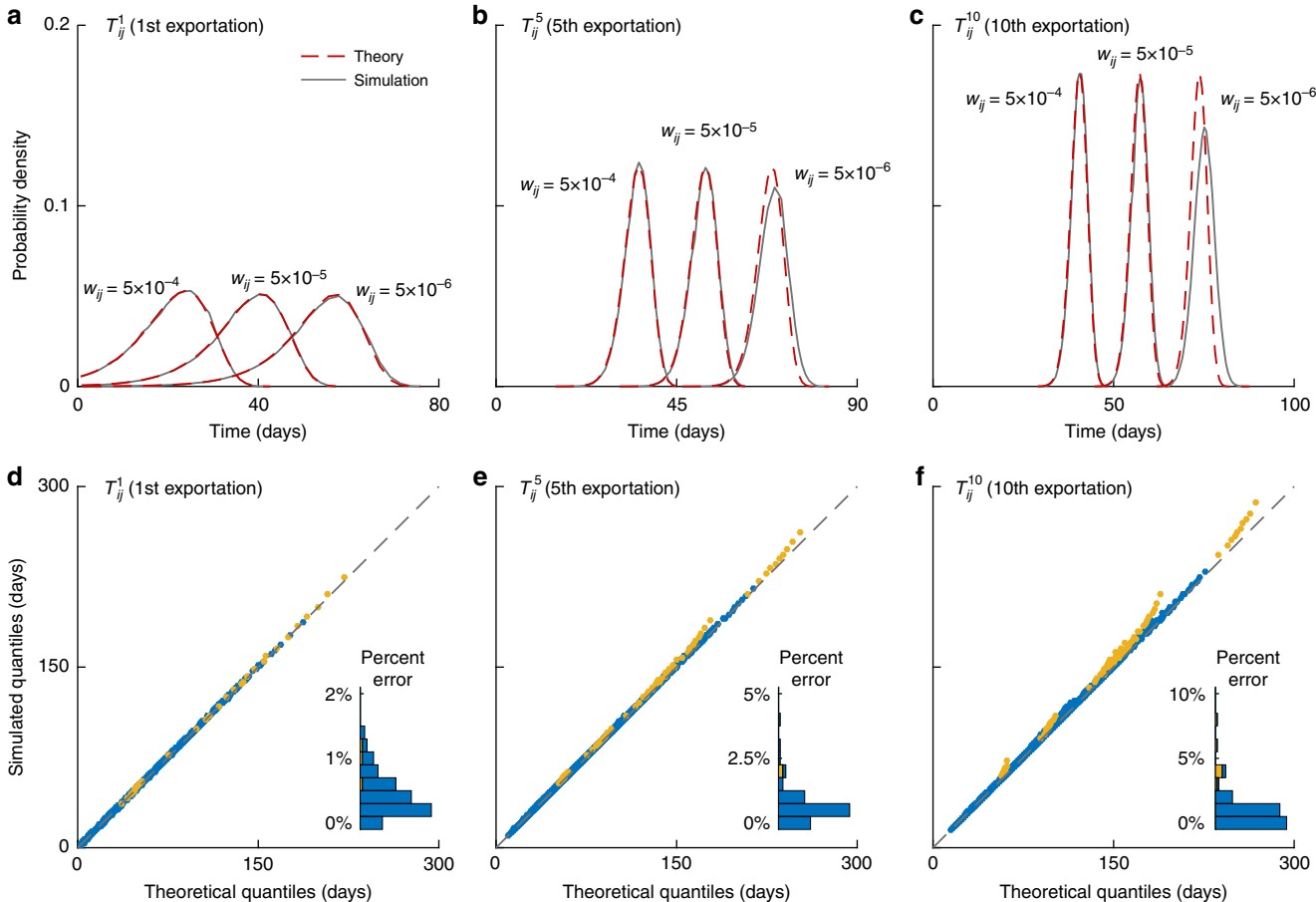

**Fig. 1** Validating the framework in the two-population model. **a–c** The analytical (red dashed lines) and simulated (gray lines) pdf of $T_{ij}^1$, $T_{ij}^5$, and $T_{ij}^{10}$ for an exemplary influenza pandemic, where the mean generation time $T_g$ is 3.5 days and the initial epidemic doubling time $t_d$ is 5 days. The epidemic origin has a population size of 7 million and is seeded with 10 infections at time 0. The mobility rate $w_{ij}$ is $5 \times 10^{-6}$, $5 \times 10^{-5}$, and $5 \times 10^{-4}$ per day, which span the realistic range for populations with 1–10 million people in the WAN (Supplementary Fig. 1). **d–f** Quantile–quantile (Q–Q) plots for the analytical and simulated quantiles of $T_{ij}^1$, $T_{ij}^5$, and $T_{ij}^{10}$ across 100 epidemic scenarios randomly generated from the following parameter space using Latin-hypercube sampling: doubling time $t_d$ and generation time $T_g$ both between 3 and 30 days, seed size $s_i$ between 1 and 100. Each epidemic scenario is coupled with a set of network parameters randomly generated with mobility rate $w_{ij}$ between $10^{-6}$ and $10^{-3}$ and population size $N_i$ between 0.1 and 10 million. Simulated quantiles in each scenario are compiled using 10,000 stochastic realizations. In the Q—Q plots, deviations from the diagonal indicate discrepancies between the analytical and simulated quantiles. Data points are colored in blue if the number of exportations is $n$ or above with probability 1, and yellow otherwise. Insets show the corresponding histograms of percent error in $E[T_{ij}^n]$

York. If the epidemic origin is a hub, the growth of local disease prevalence can be substantially reduced if a significant proportion of infections travel outward as the epidemic unfolds[18]. Let $D_{i,c}$ be the set of populations that are $c$ degrees of separation from the epidemic origin in the WAN-SPT[24]. From the perspective of the importation process for a given population $j \in D_{i,1}$ (i.e., directly connected to the epidemic origin), the prevalence in population $i$ grows exponentially at rate $\lambda_{ij} = \lambda_i - \sum_{k \neq j} w_{ik}$ (see Fig. 2a and The WAN-SPT analysis in Methods for details). Figure 2d and Supplementary Fig. 3 show that with this hub-effect adjustment, $f_n(t|\lambda_{ij}, \alpha_{ij})$ accurately characterizes the probability distribution of $T_{ij}^n$ for all populations in $D_{i,1}$ (e.g., the percent error in expected EAT is uniformly below 4%).

Continuous seeding: Unlike the epidemic origin which has a single seeding event at time 0, all the other populations in the WAN-SPT are continuously seeded by infections coming from their upstream populations. Suppose population $k \in D_{i,2}$ is connected to the epidemic origin via population $j$ along the path $\psi$: $i \to j \to k$. After the epidemic has arrived at population $j$ at time $T_{ij}^1$, population $i$ continues to export infections to population $j$ before the epidemic arrives at population $k$ at time $T_{ik}^1$ (illustrated in Fig. 2b). Under assumption 2, each imported infection in population $j$ (arriving at times $T_{ij}^1$, $T_{ij}^2$, …) spawns an infection tree that grows exponentially at the hub-adjusted rate $\lambda_{jk}$. Therefore, the prevalence in population $j$, namely $I_j(t)$, is simply the sum of the prevalence for all these infection trees. As such, assumption 1 warrants that the exportation of infections from population $j$ to $k$ is an NPP with intensity function $w_{jk}I_j(t)$, which is itself a stochastic process because of its dependence on the random variables $T_{ij}^1$, $T_{ij}^2$, … (see The WAN-SPT analysis in Methods). We conjecture that this highly complex stochastic process can be greatly simplified with little loss of accuracy by assuming that conditional on $T_{ij}^1$ (the EAT for population $j$), $T_{ij}^m \approx E\left[T_{ij}^m \mid T_{ij}^1\right]$ for all $m > 1$ (see Eq. 2). In other words, the major source of stochasticity in $I_j(t)$ comes from $T_{ij}^1$, which is characterized by $f_1(t|\lambda_{ij}, \alpha_{ij})$ (Eq. 1). Figure 2e and Supplementary Fig. 4 show that our conjecture is valid. The resulting approximate pdf of $T_{ij}^n$ is accurate for $D_{i,2}$ populations for all realistic epidemic scenarios. Furthermore, this pdf can in turn be well approximated with $f_1(t|\lambda_\psi, \alpha_\psi)$, where $\lambda_\psi$ and $\alpha_\psi$ are obtained by minimizing the relative entropy[25] (see The WAN-SPT analysis in Methods). This implies that the spread of epidemics from the

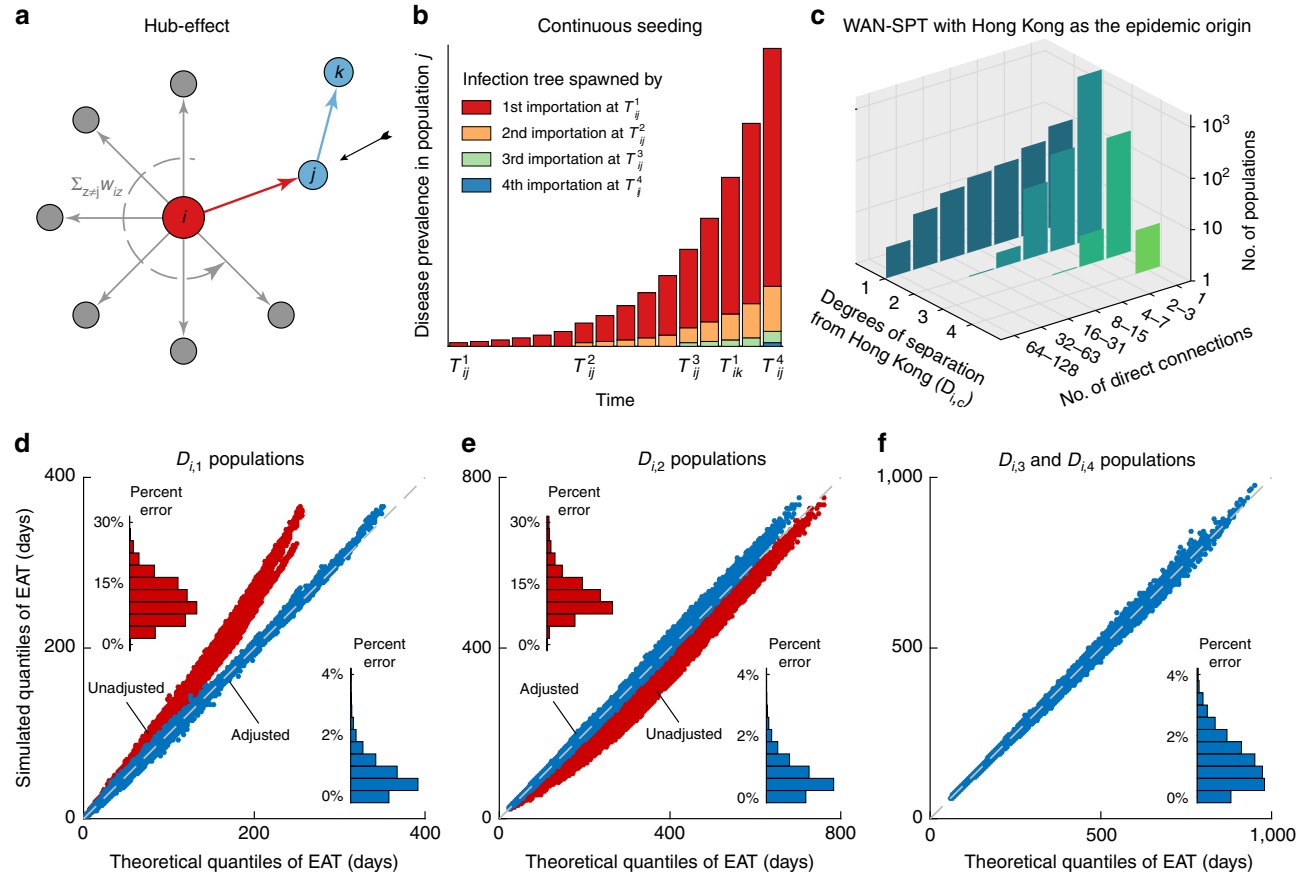

**Fig. 2** Validating the framework in the WAN-SPT. **a**, **b** Schema of the hub-effect and continuous seeding. In this example, the epidemic arrives at population $k$ after population $j$ has imported three infections from the epidemic origin, i.e., $T_{ij}^3 < T_{ik}^1 < T_{ij}^4$. In the absence of continuous-seeding adjustment, infection trees spawned by the second and subsequent importations in population $j$ are ignored[18]. **c** Basic network properties of the WAN-SPT with Hong Kong as the epidemic origin (WAN-SPT-HK). **d**–**f** Q–Q plots for the analytical and simulated quantiles of EATs for all 2308 populations in the WAN-SPT-HK across all 100 epidemic scenarios considered in Fig. 1 (i.e., 230,800 Q—Q plots in total). Insets show the corresponding histograms of percent error in expected EAT. **d** EATs for all 246 populations in $D_{i,1}$ before (red) and after (blue) adjusting for the hub-effect. **e** EATs for all 1828 populations in $D_{i,2}$ before (red) and after (blue) adjusting for continuous-seeding and path reduction; hub-effect has been adjusted for the epidemic origin and all populations in $D_{i,1}$. **f** EATs for the remaining 234 populations in $D_{i,3}$ and $D_{i,4}$ after adjusting for the hub-effect, continuous seeding and path reduction. Supplementary Figures 3–5 provide analogous results for the WAN-SPT with other major hubs as the epidemic origin

origin to any population $k \in D_{i,2}$ can be regarded as a two-population model, in which the adjusted mobility rate is $\alpha_\psi$ and the epidemic in the origin grows exponentially at rate $\lambda_\psi$. We term this procedure path reduction. By induction, we can recursively apply path reduction to characterize the EATs with comparable accuracy for all populations in $D_{i,3}$, $D_{i,4}$, etc. Figure 2f and Supplementary Fig. 5 verify this claim (e.g., the percent error in expected EAT is uniformly below 4%).

**WAN**. The accuracy of our framework for the WAN-SPT implies that for each (acyclic) path $\psi$ connecting an arbitrary population $k$ to the epidemic origin, the epidemic arrival process for population $k$ along this path can be approximated as an NPP with intensity function $\alpha_\psi \exp(\lambda_\psi t)$. In the entire WAN, each population may be connected to the epidemic origin via multiple paths (hence the difference in EATs between WAN-SPT and WAN, as shown in Supplementary Fig. 6), some of which may intersect and are therefore dependent. We conjecture that the dependence among such paths is sufficiently weak such that the overall epidemic arrival process for any population $k$ is well approximated by the superposition of the NPPs[22] that correspond to these pseudo-independent paths. That is, if $\Psi_{ik}$ is the set of all acyclic paths connecting population $k$ to the epidemic origin, the

epidemic arrival process for population $k$ can be well approximated by an NPP with intensity function $\sum_{\psi \in \Psi_{ik}} \alpha_\psi \exp(\lambda_\psi t)$. Figure 3 and Supplementary Fig. 7 show that our framework is accurate for all populations and epidemic scenarios.

**Public health applications**. Our framework provides both analytical and computational advancements for studying global spread of epidemics. First, not only can our framework be easily used to forecast EATs for all populations in the WAN, but it also analytically elucidates the dependence of EATs on the epidemiologic parameters (growth rate and seed size) and the network properties of the WAN (air-traffic volume and connectivity). Second, our framework provides closed-form probability distributions (Eq. 1) to support likelihood-based inference of key epidemiologic parameters from surveillance data on global and local spread. We exemplify the public health applications of our framework by retrospectively applying it to the 2009 influenza pandemic and the 2014 Ebola epidemic as follows.

In our first case study, we infer the transmissibility of the 2009 pandemic influenza A/H1N1 virus in Greater Mexico City following the formulation in Balcan et al.[26] Shortly after the pandemic influenza A/H1N1 virus was first detected in the USA and Mexico in April 2009, many countries enhanced their

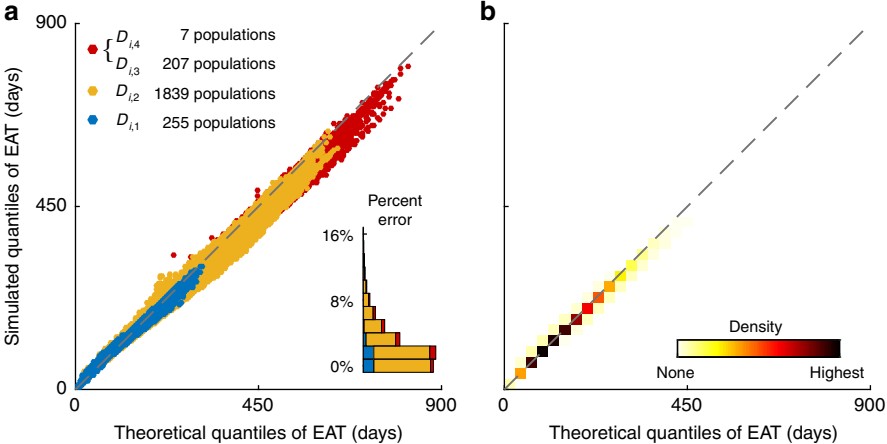

**Fig. 3** Validating the framework in the WAN. The epidemic origin is Hong Kong as in Fig. 2. **a** Analogous to Fig. 2d–f, with populations in $D_{i,c}$ ($c = 1, 2, 3, 4$) color-coded. Analytical EATs are computed using NPP superposition as described in the main text (see The WAN analysis in Methods for algorithmic details). **b** Density of the data points in **a** to show that nearly all the 230,800 Q—Q plots align with the diagonal, which indicates congruence between simulated and analytical EATs. Supplementary Figure 7 provides analogous results with other major hubs in the WAN as the epidemic origin

surveillance to monitor importations of pandemic infections. As such, data on EATs for these countries were deemed more reliable than epidemic curve data, which are typically confounded by reporting behavior and surveillance capacity[26–28]. Using GLEAM simulations powered by supercomputers to perform maximum-likelihood analyses of EATs for 12 countries seeded by Mexico, Balcan et al.[26] estimated that if the 2009 influenza pandemic started in La Gloria on 11, 18, or 25 February 2009, the basic reproductive number $R_0$ would be 1.65 (1.54–1.77), 1.75 (1.64–1.88), or 1.89 (1.77–2.01), respectively (Fig. 4a). Integrating our framework into their inference formulation, we can express the likelihood as a simple analytical function of $R_0$ (see case study on the 2009 influenza A/H1N1 pandemic in Methods) and obtain essentially the same $R_0$ estimates without the need for super-computing (Fig. 4a). Specifically, our point estimate of $R_0$ would be the same as that in Balcan et al. if the epidemic in Greater Mexico City began with a single seed on 22 February, 1 March, or 9 March 2009, respectively, which are all consistent with range of the epidemic start times documented in surveillance reports[29] and other studies[27,28,30,31]. The reduction in computational complexity and requirement provided by our framework translates into substantial improvement for timeliness and efficiency in situational awareness.

In our second case study, we analyze the 2014 West African Ebola epidemic in Montserrado and Margibi, Liberia (Montserrado henceforth for brevity). Specifically, we apply our framework to retrospectively nowcast the reporting proportion of Ebola cases (and hence the total number of cases) in Montserrado, and forecast the time to the next international case exportation from Montserrado assuming that the local epidemic would continue to grow exponentially at the nowcasted growth rate and the forward air-traffic would remain constant.

The Montserrado Ebola epidemic started in May 2014[32,33]. By September 2014, two indigenous Ebola cases had been exported from Montserrado to other nations via commercial air travel: The first to Lagos, Nigeria, on 20 July 2014[34]; and the second to Dallas, USA, on 19 September 2014[35]. By combining these global spread data with the World Health Organization patient database[33] on the weekly number of confirmed and probable cases in Montserrado and accounting for the effect of the opening of new Ebola treatment units in August[36,37], we can use our framework to express the likelihood function in simple analytical form (see case study on the 2014 Liberian Ebola outbreak in Methods). We estimate that the reporting proportion (and hence

the total number of cases) would have been statistically identifiable starting from 6 July 2014 onwards. We estimate that by 6 July 2014, the confirmed and probable cases only accounted for 18% (95% credible interval 7–33%) of all Ebola cases in Montserrado. The opening of new treatment units during August increased the reporting proportion to 30% (15–48%) by 17 August 2014, which is congruent with an independent estimate[38] based on capture–recapture sampling of raw patient records over a similar time horizon (34%; 95% confidence interval 26–50%). Retrospective real-time forecasts of the time to next exportation are consistent with the observed exportation times (namely 20 July and 19 September) except on 21–28 July during which the next exportation occurred later than predicted. The prediction errors on 21–28 July could be attributed to travel restrictions started in August[8], the effect of which could not be included in the forecasts until they have actually occurred during August. If travel restrictions could have been foreseen on 21–28 July and incorporated into the forecasts (as a counterfactual scenario for illustration), the observed case exportation on 19 September 2014 would be consistent with the forecast range (Fig. 4b). These conclusions are robust against temporal variations in epidemic growth rate (see case study on the 2014 Liberian Ebola outbreak in Methods).

## Discussion

In summary, our framework for characterizing the dynamics underlying global spread of epidemics comprises five approximations: (i) a closed-form pdf for EAT for any two directly connected populations (Eq. 1); (ii) adjustment for hub-effects; (iii) adjustment for continuous seeding; (iv) path reduction; and (v) path superposition. Approximation (i) is the indispensable centerpiece of our framework, whereas the necessity of approximations (ii)–(v) would depend on the specific application. Hub-effect adjustment is necessary when estimating the times of case exportation for populations that are directly connected to multiple populations and have relatively high outbound mobility rates. Continuous-seeding adjustment is necessary when estimating the times of case exportation for all populations except the epidemic origin (for which seeding is assumed to occur only at time 0). Path reduction and superposition are developed for simplifying computation as well as generating insights regarding global spread dynamics. In terms of computation, path reduction is required for populations that are three or more degrees of separation from the epidemic origin in a given acyclic path

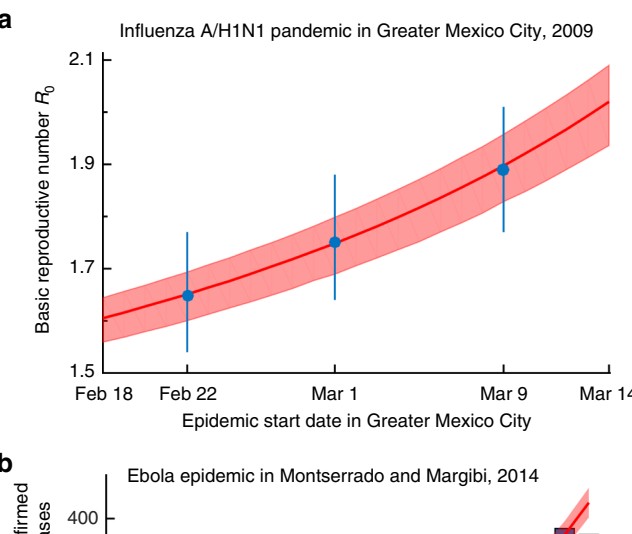

**a**

Influenza A/H1N1 pandemic in Greater Mexico City, 2009

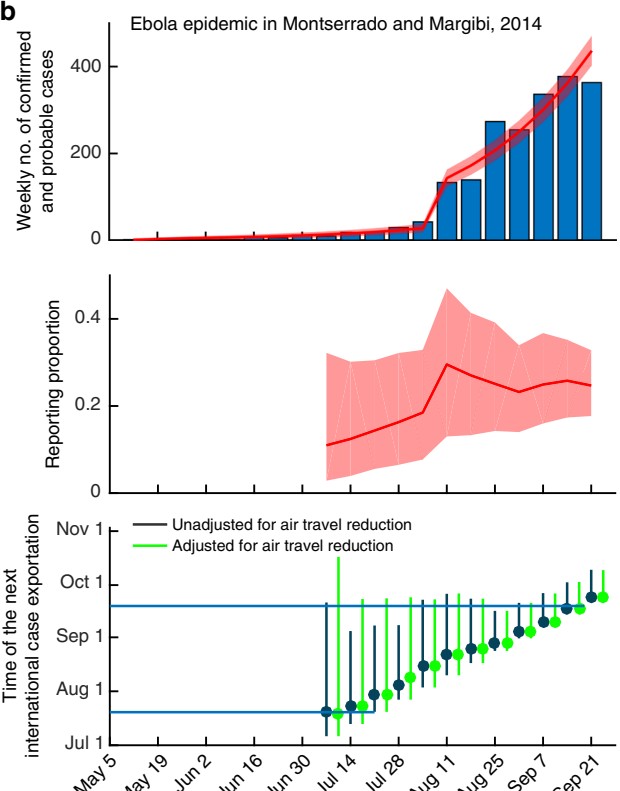

**b**

Ebola epidemic in Montserrado and Margibi, 2014

**Fig. 4** Inferring key epidemiologic parameters from surveillance data on global and local spread. Red lines and shades indicate posterior medians and 95% credible intervals of parameter estimates, respectively. **a** Case study of the 2009 influenza A/H1N1 pandemic in Greater Mexico City. The basic reproductive number $R_0$ is inferred from the observed EATs for the 12 countries seeded by Mexico as formulated in Balcan et al.[26] Blue circles and error bars indicate the $R_0$ estimates and their 95% confidence intervals in Balcan et al. assuming that the pandemic started in La Gloria on 11, 18, or 25 February 2009. **b** Case study of the 2014 Ebola epidemic in Montserrado and Margibi, Liberia. The top panel shows the weekly number of confirmed and probable Ebola cases (bars) and the fitted epidemic curve based on parameters estimated from surveillance data up to 21 September 2014. The middle panel shows retrospective real-time estimates (i.e., nowcasting) of reporting proportion, where the x-axis indicates the date of inference. The bottom panel shows retrospective real-time forecasts of the time to the next international case exportation, with and without adjusting for air travel restrictions started in August 2014. Circles and bars indicate the medians and 99% range of forecasts, respectively. Blue horizontal lines indicate the international case exportations occurred on 20 July and 19 September, 2014. Methods and Supplementary Fig. 10 provide more details and sensitivity analysis

(which typically account for <20% of all populations), whereas path superposition is used for all populations (however, path superposition may not be necessary for populations that are directly connected to the epidemic origin with high mobility rates because the indirect paths have only minor impact on their EATs; see Supplementary Fig. 6). In terms of insights, the accuracy of path reduction implies that the epidemic arrival process from the epidemic origin to any given population along any given acyclic path $\psi$ can be accurately approximated as an NPP with intensity function $\alpha_\psi \exp(\lambda_\psi t)$, whereas the accuracy of path superposition implies that the dependence of multiple paths connecting a given population to the epidemic origin is relatively weak for the purpose of estimating EAT.

Although approximation (ii)–(v) are all necessary for estimating EAT in the WAN (Figs. 1–3), they are not needed in our case studies on inference of transmission parameters: In the 2009 pandemic influenza A/H1N1 case study, we follow the inference formulation in Balcan et al.[26] which included only populations that are directly connected to Mexico City in the WAN. In the 2014 Ebola case study, the inference formulation tracks the timing of only two case exportations without the need to stratify them by outbound populations (see Case study on the 2014 Liberian Ebola outbreak in Methods).

Our study has several limitations. First, we did not consider age structure because the OAG air-traffic data do not have age information. If data are available for stratifying mobility rates and incidence by age, our framework should remain valid if the mobility rates $w_{ij}$ are calculated as the cross-product of age-specific mobility rates and age distribution of the disease. Second, we have assumed that each imported case spawns an exponentially growing infection tree with probability 1, whereas if we account for stochasticity in transmission dynamics, each imported case will fail to spawn an exponentially growing infection tree with probability $p = 1/R_0$[16]. Because this effect is similar to that of border control[19], we conjecture that our framework can be extended to account for such stochasticity in transmission dynamics by discounting $w_{ij}$ with $1 - p$. Third, we present our framework in the context of the classic SIR model. Nonetheless, our results can be generalized to all $SE_mI_nR$ models[39] (see Generalizing to $SE_mI_nR$ models in Methods). Fourth, we have not accounted for seasonality effects which may be strong and geographically heterogeneous for diseases such as seasonal influenza[40,41]. Although the epidemic dynamics will certainly be less analytically tractable in the presence of seasonality (e.g., the pdf of the EAT can no longer be well approximated by simple closed-form expressions as Eq. 1), we conjecture that the new analytics introduced here, namely adjustments for the hub-effect and continuous seeding as well as path reduction and superposition, will be useful for building a more general framework for global spread of epidemics. Finally, in our case studies, we have implicitly assumed that surveillance data were available in near real-time for nowcasting and forecasting, whereas in reality the availability of reliable data would likely incur longer lead times, and hence the timeliness of situational awareness implied here should be interpreted within such context.

In summary, we have developed a novel framework that can accurately characterize how global spread of epidemics depends on the infectious disease epidemiology and network properties of the WAN. Together with state-of-the-art global epidemic simulators such as GLEAM, our framework advances the frontiers of the next-generation informatics for pandemic preparedness and responses.

## Methods

**WAN metapopulation epidemic model.** Let $S_j(t)$, $I_j(t)$, and $R_j(t)$ be the number of susceptible, infected and removed individuals in population $j$ at time $t$. Suppose $R_{0,j}$

is the basic reproductive number and $T_{g,j}$ is the mean generation time in population $j$. Let $\beta_j = R_{0,j}/T_{g,j}$ be the disease transmission rate in population $j$, and $w_{jk}$ be the mobility rate from population $j$ to $k$. The stochastic metapopulation model with $G$ populations is specified by the following equations where $\Delta t$ is a very small time interval:

$$S_j(t+\Delta t) = S_j(t) - \underbrace{U_j(t)}_{\substack{\text{No. of infections in} \\ \text{population } j \text{ between} \\ \text{time } t \text{ and } t+\Delta t}} + \sum_k \underbrace{X_{kj}(t)}_{\substack{\text{No. of susceptibles who} \\ \text{travel from population } k \\ \text{to population } j \text{ between} \\ \text{time } t \text{ and } t+\Delta t}}$$

$$- \sum_k \underbrace{X_{jk}(t)}_{\substack{\text{No. of susceptibles who} \\ \text{travel from population } j \\ \text{to population } k \text{ between} \\ \text{time } t \text{ and } t+\Delta t}}$$

$$I_j(t+\Delta t) = I_j(t) + \underbrace{U_j(t)}_{\substack{\text{No. of infections in} \\ \text{population } j \text{ between} \\ \text{time } t \text{ and } t+\Delta t}} - \underbrace{V_j(t)}_{\substack{\text{No. of infected cases who} \\ \text{recover or die in population } j \\ \text{between time } t \text{ and } t+\Delta t}}$$

$$+ \sum_k \underbrace{Y_{kj}(t)}_{\substack{\text{No. of infected cases who} \\ \text{travel from population } k \\ \text{to population } j \text{ between} \\ \text{time } t \text{ and } t+\Delta t}} - \sum_k \underbrace{Y_{jk}(t)}_{\substack{\text{No. of infected cases who} \\ \text{travel from population } j \\ \text{to population } k \text{ between} \\ \text{time } t \text{ and } t+\Delta t}}$$

$$R_j(t+\Delta t) = R_j(t) + \underbrace{V_j(t)}_{\substack{\text{No. of infected cases who} \\ \text{recover or die in population } j \\ \text{between time } t \text{ and } t+\Delta t}}$$

$$+ \sum_k \underbrace{Z_{kj}(t)}_{\substack{\text{No. of recovered cases who} \\ \text{travel from population } k \\ \text{to population } j \text{ between} \\ \text{time } t \text{ and } t+\Delta t}} - \sum_k \underbrace{Z_{jk}(t)}_{\substack{\text{No. of recovered cases who} \\ \text{travel from population } j \\ \text{to population } k \text{ between} \\ \text{time } t \text{ and } t+\Delta t}}$$

$$U_j(t) = \beta_j S_j(t) I_j(t) \Delta t / N_j$$

$$V_j(t) = I_j(t) \Delta t / T_{g,j}$$

$$X_j(t) \sim \text{Multinomial}\left(\lfloor S_j(t) \rfloor, w_{j1}\Delta t, \dots, w_{jG}\Delta t\right)$$

$$Y_j(t) \sim \text{Multinomial}\left(\lfloor I_j(t) \rfloor, w_{j1}\Delta t, \dots, w_{jG}\Delta t\right)$$

$$Z_j(t) \sim \text{Multinomial}\left(\lfloor R_j(t) \rfloor, w_{j1}\Delta t, \dots, w_{jG}\Delta t\right),$$

where $X_{jk}(t), Y_{jk}(t)$, and $Z_{jk}(t)$ are the $k$th component of $X_j(t)$, $Y_j(t)$, and $Z_j(t)$, respectively. Multinomial($n$, $p_1,...,p_G$) denotes a multinomial random variable with $n$ trials and probabilities $p_1,...,p_G$. We use $\Delta t = 0.05$ days in all of our simulations.

**The global epidemic simulator.** We build the global simulator using 2015 worldwide flight booking data from the Official Airline Guide (OAG, https://www.oag.com/) and the Gridded Population of the World Version 4 (GPWv4, http://sedac.ciesin.columbia.edu/data/collection/gpw-v4/) data set from the NASA Socioeconomic Data and Applications Center (SEDAC) at Columbia University.

Worldwide air-transportation data: Our OAG worldwide flight booking data set contains all air bookings that have taken place in all commercial airports worldwide during 2015. Each data record contains the following information for a flight route: (i) origin airport, (ii) destination airport, (iii) connecting airports (if any), and (iv) passenger bookings for each month. The city and country served by each airport and the coordinates of each airport are known. The raw data comprises 0.947 million records. Parameterizing the WAN using these raw data would therefore generate 0.947 million connections in the network, which is beyond our computational capacity and unnecessary for an accurate description of global spread (because the WAN is densely connected)[3–6,12,42–44]. As such, we perform the following steps to exclude flight routes with weak traffic from the WAN without compromising the realism of the global epidemic simulator:

1. We exclude all routes with no bookings for one or more months during 2015.
2. We exclude all routes in which the origin or destination is a remote area with very small population size (e.g., hamlets, settlements, or communities in Alaska and Northern Canada).
3. We exclude all routes with strong seasonality as measured by normalized information entropy[11]: $H_{ij} = -\frac{1}{\log(12)}\sum_{m=1}^{12} \rho_{ijm} \log \rho_{ijm}$, where $\rho_{ijm} = F_{ijm} / \sum_{m=1}^{12} F_{ijm}$ and $F_{ijm}$ denotes the number of air bookings from origin airport $i$ to destination airport $j$ in month $m$. The measure $H_{ij}$ ranges between 0 and 1, and decreases as temporal variation in air-traffic increases (e.g., if air-traffic is the same across all months, then $H_{ij} = 1$). On the basis of the distribution of $H_{ij}$ in our OAG raw data, we exclude all routes with $H_{ij} < 0.8$ (Supplementary Fig. 8a).

Global population data: The GPWv4 data set integrates the highest resolution census data from the 2010 round of Population and Housing Censuses collected from hundreds of national statistics departments and organizations[45,46]. GPWv4 provides eight different data sets, most of which are specialized geospatial metadata that partition the global population into a grid of cells with resolution of 30 arc-second (~1 km at the equator). We use the vector data set "Administrative Unit Center Points with Population Estimates, v4 (2000, 2005, 2010, 2015, 2020)"[47], because it provides all the information that we need to build the global epidemic model, e.g., the coordinates of centroid are available for each of the ~12.5 million administrative census units (ACUs).

The WAN model: We combine our OAG data with the GPWv4 data to calculate the population size of the catchment area of each airport as follows:

1. We use the coordinates of the centroids of all ACUs and airports to calculate the great circle distance for all possible combinations of ACUs and airports within the same country. We use a Voronoi-like tessellation algorithm proposed by Balcan et al.[23] to link each ACU to its serving airport (i.e., the closest airport in its country). In this algorithm, we impose the constraint that the great circle distance between any pair of ACU and airport cannot exceed 200 km, according to the distribution of great circle distance for all combinations of ACUs and airports (Supplementary Fig. 8b). This reflects a reasonable upper bound on the distance of land transportation for reaching an airport[23]. Without this constraint, the algorithm may generate unreasonably large catchment areas for airports located in sparsely populated regions. Among the 7,995,985 ACUs with human habitats, only 45,692 are excluded from our model because of this constraint. The total population size served by an airport is the sum of populations for all ACUs assigned to that airport.
2. To strike a balance between computational requirement (within our capacity) and realism of our global epidemic simulator, we exclude all routes having less than 3000 air passengers throughout the year (Fig. S8c–f). This simplification is in line with the passenger threshold reported by Khan et al.[48] and hence has little impact on the accuracy of global spread dynamics.
3. In our OAG data set, some metropolitans (e.g., London, New York City, and Shanghai) and tourist locations (e.g., Hawaii and Canary Islands) have multiple airports. We model each of these locations as a single population by merging its serving airports and the corresponding catchment areas.
4. The daily air-traffic of each connection $F_{ij}$ is the average number of air passengers per day for that connection during the year of 2015. The ensemble of all connections shows a high degree of statistical symmetry, $F_{ij} \approx F_{ji}$ ($R^2 = 0.9981$), as in refs. [3–7,11,18,20,42–44]. As such, we symmetrize the air-traffic between each pair of populations by setting $F_{ij} = F_{ji} = (F_{ij} + F_{ji})/2$.

In summary, the WAN in our global metapopulation epidemic model comprises 54,106 connections and 2309 populations and preserves more than 92% of the global air bookings.

**Details on assumption 1.** Assumption 1 is stated as follows: suppose populations $j$ and $k$ are directly connected in the WAN and only population $j$ is infected. Exportation of infections from population $j$ to $k$ is an NPP[22] with intensity function $w_{jk}I_j(t)$ where $I_j(t)$ is the disease prevalence in population $j$ at time $t$. Previous studies[19,21] on global spread have made similar assumptions.

A counting process $\{A(t), t \geq 0\}$, where $A(t)$ is the number of events by time $t$, is an NPP[22] with intensity function $\mu(t)$ for some small time interval $\Delta t$ if:

1. $A(0) = 0$.
2. Non-overlapping increments are independent, i.e., $A(T_2) - A(T_1)$ and $A(T_4) - A(T_3)$ are independent if the time intervals $[T_1, T_2]$ and $[T_3, T_4]$ do not overlap.
3. $P(A(t+\Delta t) - A(t) = 1) = \mu(t)\Delta t + o(\Delta t)$ and $P(A(t+\Delta t) - A(t) > 1) = o(\Delta t)$ for all $t$ and $o(\Delta t)/\Delta t \to 0$ as $\Delta t \to 0$.

For populations $j$ and $k$ mentioned above, the exportation process of infections from population $j$ to population $k$ clearly satisfies conditions 1 with intensity function $w_{jk}I_j(t)$. If the mobility rate $w_{jk}$ is sufficiently small, the number of exportations is only a very small proportion of the disease prevalence in population $j$, and hence conditions 2 and 3 are also satisfied.

**The two-population model analysis.** Population $i$ is the epidemic origin and only connected to population $j$. Let $s_i$ and $\lambda_i$ be the seed size and the initial epidemic growth rate. Let $X_{ij}$ be the total number of infections imported by population $j$ over

the course of the epidemic. We denote the Poisson pdf at value $x$ with mean $M$ by $f_{Poisson}(x, M)$. Under assumption 1:

1. $X_{ij}$ is Poisson distributed with mean $A_i T_g F_{ij}$, where $T_g$ is the mean generation time, $A_i$ is the final attack rate in population $i$ and $F_{ij}$ is the daily average number of passengers traveling from population $i$ to $j$. That is, $P(X_{ij} = n) = f_{Poisson}(n, A_i T_g F_{ij})$.
2. Applying the framework of NPP[22], we express the pdf of $T_{ij}^n$ conditional on $X_{ij} \geq n$ as

$$\frac{f_{Poisson}\left(n-1, w_{ij}\int_0^t I_i(u)du\right)}{P(X_{ij} \geq n)} w_{ij} I_i(t). \tag{S1}$$

Supplementary Figure 2 shows that the pdf in Eq. S1 is very accurate for all realistic epidemic scenarios. If assumption 2 is also valid, i.e., $I_i(t) = s_i \exp(\lambda_i t)$, then $P(X_{ij} \geq n) = 1$ and Eq. S1 can be simplified to

$$f_n(t|\lambda_i, \alpha_{ij}) = \left(\frac{\exp(\lambda_i t)-1}{\lambda_i}\right)^{n-1} \frac{\alpha_{ij}^n}{(n-1)!} \exp\left[\lambda_i t - \frac{\alpha_{ij}}{\lambda_i}(\exp(\lambda_i t)-1)\right],$$

which is Eq. 1 in the main text with $\alpha_{ij} = s_i w_{ij}$. The corresponding cumulative distribution function (cdf) is given by

$$F_n(t|\lambda_i, \alpha_{ij}) = \Gamma\left(n, \frac{\alpha_{ij}}{\lambda_i}(\exp(\lambda_i t)-1)\right),$$

where $\Gamma$ is the lower incomplete gamma function. The expected EAT is given by

$$E\left[T_{ij}^1\right] = \frac{1}{\lambda_i} \exp\left(\frac{\alpha_{ij}}{\lambda_i}\right) E_1\left(\frac{\alpha_{ij}}{\lambda_i}\right),$$

where $E_m(x) = x^{m-1} \int_x^\infty \frac{\exp(-u)}{u^m} du$ is the exponential integral.

If $\alpha_{ij} \ll \lambda_i$ and $\gamma$ denotes the Euler constant, we obtain the following approximation

$$E\left[T_{ij}^1\right] \approx \frac{1}{\lambda_i} \left[\ln\left(\frac{\lambda_i}{\alpha_{ij}}\right) - \gamma\right],$$

which is congruent with the EAT statistic in Gautreau et al.[18] for estimating the order of epidemic arrival across different populations.

The expected time of the $n$th exportation is given by

$$E[T_{ij}^n] = \frac{1}{\lambda_i} \exp\left(\frac{\alpha_{ij}}{\lambda_i}\right) \sum_{m=1}^n E_m\left(\frac{\alpha_{ij}}{\lambda_i}\right).$$

For any positive integers $m$ and $n$ such that $m < n$, the pdf of $T_{ij}^n - T_{ij}^m$ conditional on $T_{ij}^m$ is simply $f_{n-m}\left(t|\lambda_i, \alpha_{ij} \exp(\lambda_i T_{ij}^m)\right)$ which corresponds to the time of the $(n-m)$th exportation for an epidemic with seed size $s_i \exp(\lambda_i T_{ij}^m)$. Using this relation recursively, we deduce that the joint pdf of $T_{ij}^1 = t_1, \ldots, T_{ij}^n = t_n$ is simply

$$\prod_{m=1}^n f_1\left(t_m|\lambda_i, \alpha_{ij}\exp(\lambda_i t_{m-1})\right) \text{ for all } 0 = t_0 < t_1 < t_2 < \ldots < t_{n-1} < t_n, \tag{S2}$$

which is the basis that supports our likelihood-based inference framework. By the same token,

$$E[T_{ij}^n|T_{ij}^1] = T_{ij}^1 + \frac{1}{\lambda_i} \exp\left(\frac{\alpha_{ij}\exp(\lambda_i T_{ij}^1)}{\lambda_i}\right) \sum_{m=1}^{n-1} E_m\left(\frac{\alpha_{ij}\exp(\lambda_i T_{ij}^1)}{\lambda_i}\right)$$

which is Eq. 2 in the main text.

**The WAN-SPT analysis**. Hub-effect: Suppose the epidemic origin (population $i$) is directly connected to one or more populations, one of which is population $j$ (as illustrated in Fig. 2a). In the deterministic version of our metapopulation epidemic model (see WAN metapopulation epidemic model in Methods), the disease prevalence in population $i$ during the exponential growth phase is well approximated by the differential equation

$$\frac{dI_i}{dt} = \lambda_i I_i - \sum_k w_{ik} I_i = \left(\lambda_i - \sum_{k\neq j} w_{ik}\right) I_i - w_{ij} I_i,$$

where the actual growth rate of the disease prevalence in population $i$ is $\lambda_i - \sum_k w_{ik}$. This differential equation leads us to make the following conjecture: In our original stochastic model, in which the epidemic arrival process for population $j$ is essentially an NPP with intensity function being the second term of the above equation (i.e., $w_{ij} I_i$), we can estimate the EAT for population $j$ using the results from the two-population model (The two-population model analysis in Methods) in which population $i$ is exporting cases to population $j$ at mobility rate

$w_{ij}$ (viewed as a stochastic process) and the disease prevalence in population $i$ is growing exponentially at rate $\lambda_{ij} = \lambda_i - \sum_{k \neq j} w_{ik}$ (viewed as a deterministic process). The hub-adjusted growth rate $\lambda_{ij}$ can be interpreted as the rate at which disease prevalence in population $i$ is growing exponentially before population $j$ imports its first case from population $i$. Note that the hub-adjusted rate $\lambda_{ij} = \lambda_i - \sum_{k \neq j} w_{ik}$ is not the same as the actual growth rate, namely $\lambda_i - \sum_k w_{ik}$. To see this, consider the two-population model in which population $i$ is only connected to population $j$. In this case, the EAT distribution is given by Eq. 1, which requires $\lambda_{ij}$ to be the hub-adjusted rate $\lambda_i - \sum_{k \neq j} w_{ik} = \lambda_i$ but not the actual growth rate $\lambda_i - \sum_k w_{ik} = \lambda_i - w_{ij}$.

Continuous seeding: Consider the path connecting the epidemic origin to population $k$ via population $j$, i.e., $\psi : i \to j \to k$. Let $\lambda_{ij}$ and $\lambda_{jk}$ be the hub-adjusted growth rate in populations $i$ and $j$ for this path. Under assumption 2, the prevalence in population $j$ at time $t$ that are spawned by the $m$th infection imported from population $i$ is $\mathbf{I}\left\{t > T_{ij}^m\right\} \exp\left(\lambda_{jk}\left(t - T_{ij}^m\right)\right)$ where $\mathbf{I}\{\cdot\}$ is the indicator function. Therefore, the total prevalence in population $j$ at time $t$ is $I_j(t) = \sum_{m=1}^\infty \mathbf{I}\left\{t > T_{ij}^m\right\} \exp\left(\lambda_{jk}\left(t - T_{ij}^m\right)\right)$. The NPP intensity function for the exportation of infections from population $j$ to population $k$ is $w_{jk} I_j(t)$. Conditional on $I_j$ and hence $T_{ij}^1, T_{ij}^2, \ldots$, the pdf of $T_{ik}^n$ is $g_n(t|w_{jk}I_j) = f_{Poisson}\left(n-1, w_{jk}\int_0^t I_j(u)du\right) w_{jk} I_j(t)$ for $n = 1, 2, \ldots$. The unconditional pdf of $T_{ik}^n$ is thus $E_{T_{ij}^1, T_{ij}^2, \ldots}\left[g_n(t|w_{jk}I_j)\right]$ where the joint pdf of $T_{ij}^1 = t_1, T_{ij}^2 = t_2, \ldots$, is simply the product of $f_1\left(t_m|\lambda_{ij}, w_{ij}s_i \exp(\lambda_{ij}t_{m-1})\right)$ for $m = 1, 2, \ldots$ (see Eq. S2). As described in the main text, we make the certainty equivalent assumption (CEA) that conditional on $T_{ij}^1$, $T_{ij}^m = E\left[T_{ij}^m|T_{ij}^1\right]$ for all $m > 1$. As such, conditional on $T_{ij}^1$, we approximate $I_j$ with

$$I_j^{CEA}(t) = \sum_{m=1}^\infty \mathbf{I}\left\{t > E\left[T_{ij}^m|T_{ij}^1\right]\right\} \exp\left(\lambda_{jk}\left(t - E\left[T_{ij}^m|T_{ij}^1\right]\right)\right)$$
$$= \sum_{m=1}^\infty \mathbf{I}\left\{t > T_{ij}^1 + \Delta T_{ij}^m\right\} \exp\left(\lambda_{jk}\left(t - T_{ij}^1 - \Delta T_{ij}^m\right)\right)$$
$$= \exp\left(\lambda_{jk}\left(t - T_{ij}^1\right)\right) \sum_{m=1}^\infty \mathbf{I}\left\{t > T_{ij}^1 + \Delta T_{ij}^m\right\} \exp\left(-\lambda_{jk}\Delta T_{ij}^m\right)$$

where $\Delta T_{ij}^m = E\left[T_{ij}^m|T_{ij}^1\right] - T_{ij}^1 = \frac{1}{\lambda_{ij}} \exp\left(\frac{w_{ij}s_i\exp(\lambda_{ij}T_{ij}^1)}{\lambda_{ij}}\right) \sum_{q=1}^{m-1} E_q\left(\frac{w_{ij}s_i\exp(\lambda_{ij}T_{ij}^1)}{\lambda_{ij}}\right)$ (see Eq. 2 and the previous section).

The resulting unconditional pdf of $T_{ik}^n$ is $E_{T_{ij}^1}\left[g_n\left(t|w_{jk}I_j^{CEA}\right)\right]$ where the pdf of $T_{ij}^1$ is $f_1(\cdot|\lambda_{ij}, s_i w_{ij})$.

Path reduction: Consider the path $\psi : i \to j \to k$ in the previous section. We can approximate the pdf $E_{T_{ij}^1}\left[g_n\left(t|w_{jk}I_j^{CEA}\right)\right]$ for $T_{ik}^n$ with $f_n(t|\lambda_\psi, \alpha_\psi)$, where $\lambda_\psi$ and $\alpha_\psi$ are obtained by minimizing the relative entropy[25] for $n = 1$ (the first exportation)

$$\int_0^\infty E_{T_{ij}^1}\left[g_1\left(t|w_{jk}I_j^{CEA}\right)\right] \ln\left(\frac{E_{T_{ij}^1}\left[g_1\left(t|w_{jk}I_j^{CEA}\right)\right]}{f_1\left(t|\lambda_\psi, \alpha_\psi\right)}\right) dt.$$

This is a simple two-dimensional optimization problem. The accuracy of such path reduction (Fig. 2f and Supplementary Fig. 5) implies that the spread of epidemics from the origin to any population $k \in D_{i,2}$ can be regarded as a two-population model, in which (i) the adjusted mobility rate is $\alpha_\psi$ and (ii) the epidemic in the origin grows exponentially at rate $\lambda_\psi$. Next, consider the path $\phi : i \to j \to k \to m$, i.e., $m \in D_{i,3}$. Using path reduction, we can approximate $\phi$ with $\phi' : i \to k \to m$ where the adjusted mobility rate and epidemic growth rate in the origin for the $i \to k$ leg are $\alpha_\psi$ and $\lambda_\psi$, respectively. The arrival times of imported cases in population $m \in D_{i,3}$ (i.e., $T_{im}^n, n = 1, 2, \ldots$) can then be estimated using the tools (i.e., adjustments for hub-effect and continuous seeding) that we have developed for $D_{i,2}$ populations. The arrival times of imported cases for population $D_{i,c}, c = 4, 5, \ldots$, can be estimated analogously.

**The WAN analysis**. Superposition of paths: Let population $i$ be the epidemic origin and consider population $k \in D_{i,c}$, i.e., population $k$ is $c$ degrees of separation from the epidemic origin[24]. Superposition of NPPs for paths connecting population $i$ to $k$ is implemented as follows. As in the main text, let $\Psi_{ik}$ be the set of all acyclic paths connecting the epidemic origin to population $k$. Enumeration of all paths in $\Psi_{ik}$ for every population in the WAN is computationally prohibitive[49] (and unnecessary). Instead, we approximate $\Psi_{ik}$ with the 25 "fastest" paths from population $i$ to $k$ that are identified using the following algorithm:

1. Use the depth-first search algorithm[49] to identify the set of acyclic paths from the epidemic origin to population $k$ that have at most $c + 2$ connections. We denote this set by $\Omega_{ik}$ and assume that all the paths not in $\Omega_{ik}$ have negligible contribution to the EAT for population $k$.
2. Define the distance between any two directly connected populations $a$ and $b$ as $-\ln(w_{ab})$, which is analogous to the distance metric in Brockmann and Helbing[20], namely $1 - \ln(w_{ab}/\sum_b w_{ab})$. We choose to use this distance metric because (as described in The two-population model analysis in

Methods) if population $j$ is directly connected to population $i$, then $E\left[T_{ij}^1\right] \approx \frac{1}{\lambda_i}\left[\ln\left(\lambda_i/\alpha_{ij}\right) - \gamma\right]$ given $\alpha_{ij} \ll \lambda_i$, where $\gamma$ denotes the Euler constant and $\alpha_{ij} = s_i w_{ij}$. This indicates that the expected EAT is proportional to $-\ln\left(w_{ij}\right)$.

3. Based on our distance metric in step 2, identify the 100 shortest paths in $\Omega_{ik}$ by sorting in an ascending order. Denote the resulting set by $\Omega_{ik}^S$.

4. For each path $\psi \in \Omega_{ik}^S$, use hub-effect adjustment, continuous-seeding adjustment and path reduction developed in the WAN-SPT analysis to calculate $\lambda_\psi$ and $\alpha_\psi$ and the corresponding expected EAT, namely $\frac{1}{\lambda_\psi} \exp\left(\frac{\alpha_\psi}{\lambda_\psi}\right) E_1\left(\frac{\alpha_\psi}{\lambda_\psi}\right)$.

5. Approximate $\Psi_{ik}$ with the 25 paths in $\Omega_{ik}^S$ that have the smallest expected EATs computed in step 4 (i.e. the 25 "fastest" paths). We choose to use the 25 fastest paths in $\Omega_{ik}^S$ to approximate $\Psi_{ik}$ because Supplementary Fig. 9 shows that the accuracy of EAT estimates would slightly worsen if we use only the 10 fastest paths in $\Omega_{ik}^S$ while there is little improvement in performance if we use the 50 fastest or all paths in $\Omega_{ik}^S$.

**Generalizing to $SE_mI_nR$ models**. In the main text, our framework is built using the SIR model within each population. In this section, we describe how to generalize our framework to $SE_mI_nR$ models[39] in which:

1. The duration of latency is gamma distributed with mean $D_E$ and $m$ subclasses (i.e., with shape $m$ and rate $b_E = m/D_E$);

2. The duration of infectiousness is gamma distributed with mean $D_I$ and $n$ subclasses (i.e., with shape $n$ and rate $b_I = n/D_I$).

For any given population, let $S(t)$ be the number of susceptible individuals, $E_i(t)$ the number of individuals in the $i$th latent subclass, and $I_j(t)$ the number of individuals in the $j$th infectious subclass. The $SE_mI_nR$ system is described by the following differential equations:

$$\frac{dS(t)}{dt} = -\beta \frac{S(t)}{N} \sum_{j=1}^n I_j(t)$$

$$\frac{dE_1(t)}{dt} = \beta \frac{S(t)}{N} \sum_{j=1}^n I_j(t) - b_E E_1(t)$$

$$\frac{dE_i(t)}{dt} = b_E(E_{i-1}(t) - E_i(t)) \text{ for } i = 2, ..., m$$

$$\frac{dI_1(t)}{dt} = b_E E_m(t) - b_I I_1(t)$$

$$\frac{dI_j(t)}{dt} = b_I(I_{j-1}(t) - I_j(t)) \text{ for } j = 2, ..., n$$

During the early stage of the epidemic (such that $S(t) \approx N$), the prevalence of latent and infectious individuals both grows exponentially at rate $\lambda$ which is the solution to the following equation[39]:

$$\lambda\left(\lambda + \frac{m}{D_E}\right)^m - \beta\left(\frac{m}{D_E}\right)^m \left(1 - \left(\frac{\lambda D_I}{n} + 1\right)^{-n}\right) = 0.$$

That is, the prevalence of latent and infectious individuals are well approximated by $\overline{E} \exp(\lambda t)$ and $\overline{I} \exp(\lambda t)$, respectively, where $\overline{E}$ and $\overline{I}$ depend on the initial conditions and parameters of the differential equation systems (the analytical expressions of $\overline{E}$ and $\overline{I}$ can be obtained by solving the linearized system with $S(t) = N$). As such, if a proportion $1 - p_E$ and $1 - p_I$ of the latent and infectious individuals refrain from air travel because of their infections, then the seed size $s_0$ in the main text is simply $p_E \overline{E} + p_I \overline{I}$.

**Case study on the 2009 influenza A/H1N1 pandemic**. As described in the main text, by integrating our framework into the inference formulation in Balcan et al.[26], we express the likelihood function for the EATs for the 12 countries seeded by Mexico (see Supplementary Table 1) as

$$L(R_0) = \prod_{j \in A} f_1(t_j|\lambda_{ij}, \alpha_{ij}) \prod_{j \in B} F_1(t_j|\lambda_{ij}, \alpha_{ij})$$

in which population $i$ (the epidemic origin) is Greater Mexico City[50] where the epidemic began in mid-February to early-March 2009[27,30], $t_j$ is the observed EAT for population $j$ which can be exact ($A$) or left-censored ($B$), $\lambda_{ij} = \lambda_i - \sum_{k \neq j} w_{ik}$ is the hub-adjusted growth rate, $\alpha_{ij}$ is the adjusted mobility rate. Because the air travel data were not reported in Balcan et al.[26], we use the air travel data published in Fraser et al.[27] in which the basic reproductive number $R_0$ was estimated from the number of confirmed cases in different countries seeded by Mexico during March–April 2009. Supplementary Table 1 shows the EAT data from Balcan et al.[26] and the air-passenger data from Fraser et al.[27]. The population size of Greater Mexico City in 2009 was 17.6 million[27]. We assume that the epidemic started with a single infected individual (i.e., $s_i = 1$) in Greater Mexico City between 18 February and 14 March 2009 based on the documentation in surveillance reports[29] and other studies[27,28,30,31] (Fig. 4a). We adopt the natural history model described in Balcan et al.[26]: (i) the mean generation time is $T_g = 3.6$ days with mean latent duration of 1.1 days; (ii) the latent and infectious duration are exponentially distributed (regardless of symptoms). Under these assumptions, the basic reproductive number is $R_0 = (1 + \lambda_i \times \text{mean latent period})(1 + \lambda_i \times \text{mean infectious period})$[51]. Balcan et al.

assumed that 67% of infections are symptomatic and 50% of symptomatic infections refrained from traveling by air. As such, we discount the mobility rates by multiplying $w_{ij}$ with $0.5 \times 0.67 = 0.335$.

In this case study, $R_0$ is the only parameter subject to inference. We assume non-informative flat prior and use the Metropolis–Hasting algorithm[52,53] to estimate the posterior distributions of $R_0$. We use five MCMC chains and initialize each chain with an $R_0$ value randomly chosen between 1 and 10. The trace plot and Geweke diagnostic indicate that each MCMC chain converges within 5000 iterations and the autocorrelation of the samples in the MCMC chain is essentially 0 when the lag is larger than 10 steps. As such, we estimate the posterior distribution of $R_0$ by running the Metropolis–Hasting algorithm for 110,000 iterations with a burn-in of 10,000 iterations and a thinning interval of 10. The Gelman–Rubin diagnostic indicates that all five chains converge to the same posterior distribution.

**Case study on the 2014 Liberian Ebola outbreak**. In 2014, the first laboratory confirmed Ebola case in Montserrado, Liberia, developed symptoms during the week of 5 May 2014[32,33]. During this Ebola epidemic, two Ebola cases were exported from Montserrado to the following populations via international commercial air travel[34,35]:

1. Lagos, Nigeria on 20 July 2014 ($t_1$);
2. Dallas, USA on 19 September 2014 ($t_2$).

Montserrado and Margibi were the major epicenter in Liberia during the 2014 West African Ebola epidemic[32,33,54], and they are served by the two contiguous Liberian commercial airports that have international flights (i.e., Roberts International Airport and Spriggs Payne Airport). In this case study, we apply our framework to estimate the reporting proportion and the total number of Ebola cases in Montserrado and Margibi (Montserrado hererafter for brevity) between 5 May 2014 (the approximate start time of this epidemic) and 21 September 2014 (the last day of the week during which the last exportation occurred). Based on ref. 54, we assume that the latent period and the incubation period were the same. We assume that infectious cases did not travel by air (due to their symptoms), and exportations comprised only air travel of latent individuals (who had not yet developed symptoms). We note that there was some evidence[55] that the case exported to Lagos had already developed symptoms when he boarded the flight. Therefore, we include this case in our main analysis but exclude him in the sensitivity analysis. Results from both analyses are essentially the same (Supplementary Fig. 10).

Let time 0 be 5 May 2014 and $T$ be 21 September 2014. We denote May, June, July, August and September 2014 by months 1 to 5, respectively. Denote the last day of month $k$ since time 0 by $\tau_k$, and the two observed times of case exportations since time 0 by $t_1$ and $t_2$, respectively. We assume that the incidence rate was (i) 0 before 5 May 2014, (ii) $i_0$ on 5 May 2014, and (iii) $i_0 \exp(\lambda t)$ thereafter, i.e., this epidemic grew exponentially at rate $\lambda$ between 5 May and 21 September 2014. The incubation period has been estimated to be gamma distributed with shape $m = 1.41$ and rate $b_E = 0.154$ (which correspond to mean 9.2 days and standard deviation 7.7 days)[54]. Hence, symptomatic cases occurred at rate

$$\text{inc}_{\text{sym}}(t) = \int_0^t i_0 \exp(\lambda u)g(t - u)du = i_0 \exp(\lambda t)\left(\frac{b_E}{\lambda + b_E}\right)^m \Gamma((\lambda + b_E)t, m),$$

where $g$ is the pdf of the incubation period, and $\Gamma$ is the lower incomplete gamma function. Accordingly, the number of new symptomatic Ebola cases in the $k$th week since time 0 was

$$Y_k(\lambda, i_0) = \int_{7(k-1)}^{7k} \text{inc}_{\text{sym}}(t)dt.$$

Let $\theta_k$ be the probability that a true case with onset in week $k$ was reported as confirmed or probable cases. New Ebola treatment units were established in Montserrado in early August 2014[36,37]. As such, we assume that $\theta_k = \theta_{\text{before}}$ if week $k$ ended before 4 August 2014, and $\theta_k = \theta_{\text{after}}$ otherwise. The likelihood for the observed number of confirmed and probable cases is

$$L_{\text{inc}}(\lambda, i_0, \theta_{\text{before}}, \theta_{\text{after}}) = \prod_k f_{\text{binomial}}(y_k|Y_k(\lambda, i_0), \theta_k),$$

where $y_k$ is the observed number of confirmed and probable cases with onset in week $k$ and $f_{\text{binomial}}$ is the binomial pdf. The observed weekly number of confirmed and probable Ebola cases in Montserrado is obtained from the World Health Organization (WHO) patient database[33].

Our OAG data set also contains the monthly number of flight bookings in 2014. Supplementary Table 2 shows the monthly outbound mobility rates from Montserrado during May–September 2014 in this OAG database. We denote the outbound mobility rate from population $i$ during month $k$ by $W_k$ (i.e., $W_k = \sum_j w_{ijk}$, where $w_{ijk}$ is the daily mobility rate from population $i$ to $j$ during month $k$). Air travel restrictions were implemented starting in August[8], which presumably resulted in a substantial proportion of canceled flight bookings (in particular for

August 2014, see Supplementary Table 2). These abnormal cancellations were not registered in the OAG database. Therefore, as an approximation, we assume that the actual mobility rate in August 2014 was the same as that in September 2014.

According to our framework, if population $i$ has seed size $s$, epidemic growth rate $\lambda$, and outbound mobility rate $w$, the probability that population $i$ has no exportation up to time $t$ is

$$1 - F_1(t|\lambda, sw) = \exp\left(-\frac{sw}{\lambda}(\exp(\lambda t) - 1)\right)$$

and the probability density that population $i$ has its first exportation at time $t$ is

$$f_1(t|\lambda, sw) = sw\exp(\lambda t) \cdot (1 - F_1(t|\lambda, sw)).$$

Given the incidence rate $i_0\exp(\lambda_i t)$, the seed size of latent infections was effectively $\overline{E} = \frac{i_0}{\lambda_i}(1 - (1 + \lambda_i/b_E)^{-m})$. To see this, consider an $SE_mI_nR$ system during the exponential growth phase with incidence rate $i_0\exp(\lambda_i t)$. The prevalence of latent individuals during this phase is well approximated by the following system:

$$\frac{dE_1(t)}{dt} = i_0\exp(\lambda t) - b_E E_1(t)$$
$$\frac{dE_i(t)}{dt} = b_E(E_{i-1}(t) - E_i(t)) \text{ for } i = 2, \ldots, m .$$

Solving these differential equations gives

$$E_i(t) = \frac{i_0\exp(\lambda t)}{b_E(1 + \lambda/b_E)^i} + \exp(-b_E t) \cdot O(t^{j-1}), \text{ for } i = 1, \ldots, m,$$

and hence the prevalence of latent individuals, namely $\sum_{i=1}^{m} E_i(t)$, is well approximated by $\overline{E}e^{\lambda t}$, where $\overline{E} = (i_0/\lambda_i)(1 - (1 + \lambda_i/b_E)^{-m})$.

Taken together, the likelihood for all observed times of case exportations is

$L_{export}(\lambda, i_0)$

$= \underbrace{\left(1 - F_1(\tau_1|\lambda, \overline{E}W_1)\right)}_{\substack{\text{No exportation up to} \\ \text{31 May 2014}}} \underbrace{\left(1 - F_1(\tau_2 - \tau_1|\lambda, \overline{E}e^{\lambda\tau_1}W_2)\right)}_{\substack{\text{No exportation up to} \\ \text{30 June 2014}}} \underbrace{f_1(t_1 - \tau_2|\lambda, \overline{E}e^{\lambda\tau_2}W_3)}_{\substack{\text{The first exportation in} \\ \text{July 2014 occurred on} \\ \text{20 July 2014}}}$

$\underbrace{\left(1 - F_1(\tau_3 - t_1|\lambda, \overline{E}e^{\lambda t_1}W_3)\right)}_{\substack{\text{No exportation between 21 July 2014} \\ \text{and 31 July 2014}}} \times \underbrace{\left(1 - F_1(\tau_4 - \tau_3|\lambda, \overline{E}e^{\lambda\tau_3}W_4)\right)}_{\substack{\text{No exportation during} \\ \text{August 2014}}}$

$\underbrace{f_1(t_2 - \tau_4|\lambda, \overline{E}e^{\lambda\tau_4}W_5)}_{\substack{\text{The first exportation in September 2014} \\ \text{occurred on 19 September 2014}}} \underbrace{\left(1 - F_1(T - t_2|\lambda, \overline{E}e^{\lambda t_2}W_5)\right)}_{\substack{\text{No exportation between} \\ \text{19 September 2014 and} \\ \text{21 September 2014}}}$

In summary, we infer $(\lambda, i_0, \theta_{before}, \theta_{after})$ using the likelihood

$$L(\lambda, i_0, \theta_{before}, \theta_{after}) = L_{inc}(\lambda, i_0, \theta_{before}, \theta_{after})L_{export}(\lambda, i_0).$$

Note that $\theta_{after}$ is defined only after 3 August 2014 and hence not inferred until then. We assume non-informative flat priors for all parameters and use Gibbs sampling[52] to estimate the posterior distributions of $(\lambda, i_0, \theta_{before}, \theta_{after})$. We use five MCMC chains and initialize each chain with a starting point randomly generated from the following ranges: $\ln(2)/\lambda$ (i.e., the doubling time) between 1 and 100 days, $i_0$ between 1 and 100, $\theta_{before}$ between 0 and 1, and $\theta_{after}$ between 0 and 1. The trace plot and Geweke diagnostic indicate that each MCMC chain converges within 100,000 iterations and the autocorrelation of the samples in the MCMC chain drops below 0.05 when the lag is larger than 2000 steps. As such, we estimate the posterior distribution of $(\lambda, i_0, \theta_{before}, \theta_{after})$ by running the Gibbs sampling for 5.5 million iterations with a burn-in of 0.5 million iterations and a thinning interval of 5000. The Gelman–Rubin diagnostic indicates that all five chains converge to the same posterior distribution.

Given an estimate of $(\lambda, i_0, \theta_{before}, \theta_{after})$, the cumulative number of symptomatic Ebola cases up to time $t$ was:

$$C(t) = i_0\int_0^t e^{\lambda t}\left(\frac{b_E}{\lambda_i + b_E}\right)^m \Gamma((\lambda_i + b_E)t, m)dt$$

and the reporting proportion up to the end of week $K$ was

$$\sum_{k=1}^{K} y_k/C(7K).$$

The nowcasted posterior estimates of the epidemic doubling time (which is simply $\ln(2)/\lambda$) and initial incidence rate ($i_0$) are temporally consistent until mid-August after which both began to increase significantly. This suggests that the epidemic growth rate might have dropped since mid-August, which is plausible in view of substantial increase in mitigation efforts and resources starting in early August[36,37]. As such, we perform a sensitivity analysis by assuming that the epidemic doubling

time changed from $D_1$ to $D_2$ starting on 4 August 2014. Supplementary Figure 10 shows that our main result, namely the estimates of reporting proportion, remain essentially the same.

For the scenario unadjusted for travel restrictions (Fig. 4b, bottom panel), the retrospective real-time forecasts of the time to next international exportation are obtained by (i) assuming that mobility rates during the forecasted time period were the same as the most current mobility rates and (ii) sampling $(\lambda, i_0, \theta_{before}, \theta_{after})$ from their posterior distributions.

**Code availability.** Code is available on request from the authors.

**Data availability.** Global population data (raw data) that support the findings of this study are available from the Gridded Population of the World Version 4 (GPWv4) database at http://sedac.ciesin.columbia.edu/data/collection/gpw-v4. Restrictions apply to the availability of the worldwide air-traffic data set from the Official Airline Guide (https://www.oag.com/), which were used under license for the current study. Source data for case studies (Fig. 4; Supplementary Fig. 10) are tabulated in Supplementary Tables 1 and 2.

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

## Acknowledgements

We thank M. Lipsitch, B. J. Cowling, J. M. Read, K. Leung, H. Choi and Y. Zhang for helpful discussions. We thank C. K. Lam for assistance in data processing and technical support. We thank the Official Airline Guide, Center for International Earth Science Information Network at Columbia University, and World Health Organization (WHO) for providing their databases. This research was conducted in part using the research computing facilities and advisory services offered by Information Technology Services, The University of Hong Kong; and was done in part on the Olympus High Performance Compute Cluster at the Pittsburgh Supercomputing Center at Carnegie Mellon University, which is supported by National Institute of General Medical Sciences Models of Infectious Disease Agent Study (MIDAS) Informatics Services Group Grant 1U24GM110707. This research was supported by Harvard Center for Communicable Disease Dynamics from the National Institute of General Medical Sciences MIDAS Initiative (Grant No. U54GM088558), Area of Excellence Scheme of the Hong Kong University Grants Committee (Grant No. AoE/M-12/06), Research Grants Council Collaborative Research Fund (Grant No. CityU8/CRF/12G), and a commissioned grant from the Health and Medical Research Fund from the Government of the Hong Kong Special Administrative Region (Grant Nos. HKS-15-E03, HKS-17-E13). The content is solely the responsibility of the authors and does not necessarily represent the official views of the National Institute of General Medical Sciences, the National Institutes of Health. The funding bodies had no role in study design, data collection and analysis, preparation of the manuscript, or the decision to publish.

## Author contributions

Both authors conceived and designed the research, developed the methods, analyzed, and interpreted the results. L.W. performed the computations for global epidemic simulations and validation of the framework. J.T.W. performed the computations in the case studies. J.T.W. drafted the manuscript.

## Additional information

**Competing interests:** The authors declare no competing financial interests.

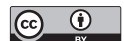

