## [Peer Review File · Nature Communications]

Reviewers' comments:

Reviewer #1 (Remarks to the Author):

The authors present an analytical framework for epidemic spread on complex transportation networks. The framework applies to fast-spreading epidemics, which allows the authors to make several simplifying assumptions. The authors apply this framework to the initial spread of pandemic H1N1 2009 and the Liberian Ebola outbreak in 2014. I believe that this contribution is sufficiently novel and useful for publication.

I am not certain that this work describes a "theory" of epidemic spread on networks. It seems to be a set of approximations that allow the authors to estimate the arrival time distribution at nodes in a reasonable network. The authors state that some of the assumptions hold for "all populations" and "epidemic scenarios" rather than stating they hold for a wide range of populations and tested scenarios. It is likely that one could construct perverse networks where the assumptions would not hold. Perhaps the manuscript's contribution is better described as a "framework" (the term used in Brockman and Helbig 2013 for the use of effective distance).

The framework makes several reasonable approximations to make it possible to compute the distribution of epidemic arrival times for each node in a network. Figures in the Supplementary Information show the effect of individual approximations. Should we assume that most or all of these assumptions are necessary, or does that depend on the application? I would like more insight about which of these approximations are most important in practice. Could any of these approximations be ignored without affecting the results of the case studies?

I request some clarification in the descriptions of the case study of pandemic H1N1. Are you assuming the epidemic started with a single individual? It is difficult to directly compare the estimate of R_0 to that from Balcan et al 2009 ($R_0=1.75$). Figure 4a plots estimated R_0 vs the start date of the epidemic, covering values from $R_0=1.6$ to $R=2.1$.

In the Discussion, on line 214, it is stated that stochastic extinction could be estimated by discounting W_{ij} . Would it be easier and more accurate to estimate the probability of non-extinction based on the number of introductions to a node? If each introduction has probability p of onward transmission, then could one estimate an epidemic start time of each node based on p and T^n ?

Reviewer #2 (Remarks to the Author):

This paper proposes and claims to validate a novel theory for how a disease will spread to different populations around the globe.

Many of the underlying assumptions, and the basis of the theory, has been established previously. In addition to improving the framing of this research with regards to the literature already cited, the relationship of the work to that of Barthelemy et al. (2010) *Journal of Theoretical Biology* 267, 554-564 should be added, and more discussion of Scalia Tomba and Wallinga (2008) could be added in particular with respect to the probability of successful invasion (in the Discussion).

It is unclear to me why the rate of import to population j is removed when considering the rate of growth in prevalence in population i (see Section 5, and bottom of page 5). Why should the growth in prevalence in a population be different dependent upon which population you are viewing it from?

I cannot completely marry the discussion of discrete-time and discrete-state dynamics (see Section 1) and initial seed of infection s_i , with continuous-time and continuous-state models as discussed, for example, in Section 5 (and in which there is a continuous flow of individuals as opposed to a seed number).

If a number of infectious individuals are assumed to arrive together, as a seed number, the probability of initial fadeout will be different to that as described in the Discussion.

I believe there is insufficient detail to be able to reliably reproduce the Path reduction mentioned as part of Sections 5 and 6. Similarly, I think the detail of the determination of the 25 "fastest" paths could be increased.

Further detail is required regarding the Markov Chain Monte Carlo inference. All parameters being inferred (or assumed known) should be detailed, along with precisely defining their prior distribution. The burn-in, convergence and mixing of chains should be discussed. What sample size has been used, and why? I also don't understand why the inferred basic reproductive number is monotonically increasing with time (this seems suspicious to me), and how this is actually consistent with the existing literature (Figure 4).

Point-by-point response (NCOMMS-17-09001)

Reviewer 1

Please note: Page and line numbers refer to those in the cleaned manuscript.

Comment 1.1

The authors present an analytical framework for epidemic spread on complex transportation networks. The framework applies to fast-spreading epidemics, which allows the authors to make several simplifying assumptions. The authors apply this framework to the initial spread of pandemic H1N1 2009 and the Liberian Ebola outbreak in 2014. I believe that this contribution is sufficiently novel and useful for publication.

Response 1.1

Thank you for your supportive comments.

Comment 1.2

I am not certain that this work describes a "theory" of epidemic spread on networks. It seems to be a set of approximations that allow the authors to estimate the arrival time distribution at nodes in a reasonable network. The authors state that some of the assumptions hold for "all populations" and "epidemic scenarios" rather than stating they hold for a wide range of populations and tested scenarios. It is likely that one could construct perverse networks where the assumptions would not hold. Perhaps the manuscript's contribution is better described as a "framework" (the term used in Brockman and Helbig 2013 for the use of effective distance).

Response 1.2

We agree that the analytics developed in this study are better described as a framework and have revised the manuscript accordingly. We have also revised the text to indicate that our results are only valid for the range of tested epidemic scenarios instead of "all populations and epidemic scenarios".

Comment 1.3

The framework makes several reasonable approximations to make it possible to compute the distribution of epidemic arrival times for each node in a network. Figures in the Supplementary Information show the effect of individual approximations. Should we assume that most or all of these assumptions are necessary, or does that depend on the application? I would like more insight about which of these approximations are most important in practice. Could any of these approximations be ignored without affecting the results of the case studies?

Response 1.3

We have added the following paragraph at the beginning of the Discussion section to address this comment:

“In summary, our framework for characterizing the dynamics underlying global spread of epidemics comprises five approximations: (i) a closed-form pdf for EAT for any two directly-connected populations (equation 1); (ii) adjustment for hub-effects; (iii) adjustment for continuous seeding; (iv) path reduction; and (v) path superposition. Approximation (i) is the indispensable centerpiece of our framework whereas the necessity of approximations (ii)-(v) would depend on the specific application. Hub-effect adjustment is necessary when estimating the times of case exportation for populations that (i) are directly connected to multiple populations and (ii) have relatively high outbound mobility rates. Continuous seeding adjustment is necessary when estimating the times of case exportation for all populations except the epidemic origin (for which seeding is assumed to occur only at time 0). Path reduction and superposition are developed for simplifying computation as well as generating insights regarding global spread dynamics. In terms of computation, path reduction is required for populations that are three or more degrees of separation from the epidemic origin in a given acyclic path (which typically account for less than 20% of all populations) while path superposition is used for all populations (however, path superposition may not be necessary for populations that are directly connected to the epidemic origin with high mobility rates because the indirect paths have only minor impact on their EATs; see Fig. S6). In terms of insights, the accuracy of path reduction implies that the epidemic arrival process from the epidemic origin to any given population along any given acyclic path ψ can be accurately approximated as an NPP with intensity function $\alpha_{\psi} \exp(\lambda_{\psi} t)$, whereas the accuracy of path superposition implies that the dependence of multiple paths connecting a given population to the epidemic origin is relatively weak for the purpose of estimating EAT.

While approximation (ii)-(v) are all necessary for estimating EAT in the WAN (Figs. 1-3), they are not needed in our case studies on inference of transmission parameters: In the 2009 pandemic influenza A/H1N1 case study, we follow the inference formulation in Balcan et al which included only populations that are directly connected to Mexico City in the WAN²⁶. In the 2014 Ebola case study, the inference formulation tracks the timing of only two case exportations without the need to stratify them by outbound populations (see Methods section 9).”

(Page 10, line 219)

Comment 1.4

I request some clarification in the descriptions of the case study of pandemic H1N1. Are you assuming the epidemic started with a single individual? It is difficult to directly compare the estimate of R_0 to that from Balcan et al 2009 ($R_0=1.75$). Figure 4a plots estimated R_0 vs the start date of the epidemic, covering values from $R_0=1.6$ to $R=2.1$.

Response 1.4

Yes, we assume that the epidemic started with a single infected individual in Greater Mexico City. We have revised the documentation of the case study as follows to address your comment:

“In our first case study, we infer the transmissibility of the 2009 pandemic influenza A/H1N1 virus in Greater Mexico City following the formulation in Balcan et al²⁶. Shortly after the pandemic influenza A/H1N1 virus was first detected in the USA and Mexico in April 2009, many countries enhanced their surveillance to monitor importations of pandemic infections. As such, data on EATs for these countries were deemed more reliable than epidemic curve data which are typically confounded by reporting behavior and surveillance capacity²⁶⁻²⁸. Using GLEAM simulations powered by supercomputers to perform maximum-likelihood analyses of EATs for 12 countries seeded by Mexico, Balcan et al.²⁶ estimated that if the 2009 influenza pandemic started in La Gloria on 11, 18, or 25 February 2009, the basic reproductive number R_0 would be 1.65 (1.54-1.77), 1.75 (1.64-1.88) or 1.89 (1.77-2.01), respectively (**Fig. 4a**). Integrating our framework into their inference formulation, we can express the likelihood as a simple analytical function of R_0 (see Methods section 8) and obtain essentially the same R_0 estimates without the need for supercomputing (**Fig. 4a**). Specifically, our point estimate of R_0 would be the same as that in Balcan et al if the epidemic in Greater Mexico City began with a single seed on 22

February, 1 March, or 9 March 2009, respectively, which are all consistent with range of the epidemic start times documented in surveillance reports²⁹ and other studies^{27,28,30,31}. The reduction in computational complexity and requirement provided by our framework translates into substantial improvement for timeliness and efficiency in situational awareness.”

(Page 8, line 169)

Fig. 4a. Case study of the 2009 influenza A/H1N1 pandemic in Greater Mexico City. The basic reproductive number R_0 is inferred from the observed EATs for the 12 countries seeded by Mexico as formulated in Balcan et al²⁶. Blue circles and error bars indicate the R_0 estimates and their 95% confidence intervals in Balcan et al assuming that the pandemic started in La Gloria on 11, 18 or 25 February 2009.

Comment 1.5

In the Discussion, on line 214, it is stated that stochastic extinction could be estimated by discounting W_{ij} . Would it be easier and more accurate to estimate the probability of non-extinction based on the number of introductions to a node? If each introduction has probability p of onward transmission, then could one estimate an epidemic start time of each node based on p and T^n ?

Response 1.5

We apologize for the confusion. We did not mean that the probability of stochastic extinction of the epidemic in a population could be estimated by discounting w_{ij} . Instead, we mean that our current framework can be extended to account for the possibility that an imported case might not spawn an exponentially growing infection tree (which occurs with probability $p = 1/R_0$) by discounting w_{ij} by $1 - p$.

We have revised the corresponding sentence as follows to clarify this.

“Second, we have assumed that each imported case spawns an exponentially growing infection tree with probability 1, whereas if we account for stochasticity in transmission dynamics, each imported case will fail to spawn an exponentially growing infection tree with probability $p = 1/R_0$ ¹⁶. Because this effect is similar to that of border control¹⁹, we conjecture that our framework can be extended to account for such stochasticity in transmission dynamics by discounting w_{ij} with $1 - p$.”

(Page 12, line 252)

Reviewer 2

Please note: Page and line numbers refer to those in the cleaned manuscript.

Comment 2.1

This paper proposes and claims to validate a novel theory for how a disease will spread to different populations around the globe.

Many of the underlying assumptions, and the basis of the theory, has been established previously. In addition to improving the framing of this research with regards to the literature already cited, the relationship of the work to that of Barthelemy et al. (2010) Journal of Theoretical Biology 267, 554-564 should be added, and more discussion of Scalia Tomba and Wallinga (2008) could be added in particular with respect to the probability of successful invasion (in the Discussion).

Response 2.1

We have followed your suggestion and added these papers in our introduction and discussion. Specifically, we have referenced “Barthelemy et al. (2010) Journal of Theoretical Biology 267, 554-564” when stating Assumption 1 (page 3, line 55) and linked our discussion on the probability of successful invasion to Scalia “Tomba and Wallinga (2008)” as follows:

“Second, we have assumed that each imported case spawns an exponentially growing infection tree with probability 1, whereas if we account for stochasticity in transmission dynamics, each imported case will fail to spawn an exponentially growing infection tree with probability $p = 1/R_0$ ¹⁶. Because this effect is similar to that of border control¹⁹, we conjecture that our framework can be extended to account for such stochasticity in transmission dynamics by discounting w_{ij} with $1 - p$.”

(Page 12, line 252)

Comment 2.2

It is unclear to me why the rate of import to population λ_j is removed when considering the rate of growth in prevalence in population λ_i (see Section 5, and bottom of page 5). Why should the growth in prevalence in a population be different dependent upon which population you are viewing it from?

Response 2.2

We have revised the description of “Hub-effect” in Section 5 of Methods as follows to clarify this point.

“*Hub effect.* Suppose the epidemic origin (population i) is directly connected to two or more populations, one of which is population j (as illustrated in **Fig. 2a**). In the deterministic version of our metapopulation epidemic model (see Section 1), the disease prevalence in population i during the exponential growth phase is well approximated by the differential equation

$$\frac{dI_i}{dt} = \lambda_i I_i - \sum_k w_{ik} I_i = \left(\lambda_i - \sum_{k \neq j} w_{ik} \right) I_i - w_{ij} I_i$$

This equation leads us to make the following conjecture: In our original stochastic model, in which the epidemic arrival process for population j is essentially an NPP with intensity function being the second term of the above equation (i.e. $w_{ij} I_i$), we can estimate the EAT for population j using the results from the two-population model (Section 4) where population i is only connected to population j with mobility rate w_{ij} and the disease prevalence in population i is growing exponentially at rate $\lambda_{ij} = \lambda_i - \sum_{k \neq j} w_{ik}$. Note that this does not mean that the disease prevalence in population i would be different dependent upon which population we are viewing it from. To further illustrate the rationale underlying this hub-effect adjustment with a simple example. Suppose population i is only connected to populations j and k and the mobility rates are $w_{ij} = 0.1\lambda_i$ and $w_{ik} = 0.01\lambda_i$. Because w_{ij} is 10 times larger than w_{ik} , the epidemic arrives in population j much earlier than population k . Because w_{ij} is not small compared to λ_i but large compared to w_{ik} , we expect that a significant percentage of disease prevalence in population i would have travelled to population j before population k imports its first case from population i . That is, the epidemic arrival process for population k is significantly slowed down by the case exportation process from population i to population j (in the sense that the former will be significantly faster if the latter is absent). This effect is approximated by our hub-effect adjustment which reduces the epidemic growth rate from λ_i to $\lambda_{ik} = \lambda_i - w_{ij} = 0.9\lambda_i$ when estimating the epidemic arrival time for population k using the two-population model. By the same reasoning, because w_{ik} is small compared to λ_i and w_{ij} , we expect that the case exportation process from population i to population k would have negligible impact on the

epidemic arrival process for population j . Indeed, the hub-adjusted growth rate for the epidemic arrival process for population j is $\lambda_{ij} = \lambda_i - w_{ik} = 0.99\lambda_i$.”

(Page 21, line 418)

Comment 2.3

I cannot completely marry the discussion of discrete-time and discrete-state dynamics (see Section 1) and initial seed of infection s_i , with continuous-time and continuous-state models as discussed, for example, in Section 5 (and in which there is a continuous flow of individuals as opposed to a seed number).

Response 2.3

We apologize for the confusion. The full model in Section 1 is continuous-time. The discrete-time formulation in Section 1 (with arbitrarily small Δt) corresponds to the numerical implementation of the full model. The initial seed of infection s_i in the epidemic origin is assumed to be an integer. The transmission dynamics within a population is deterministic with continuous state space, whereas the movements of individuals between populations is stochastic and integer-valued. We have revised Section 5 (Page 21, line 418) to clarify that the differential equation therein is used as the clue for developing the hub-effect adjustment but does not reflect the actual (stochastic) dynamics in the full model (see Response 2.2).

Comment 2.4

If a number of infectious individuals are assumed to arrive together, as a seed number, the probability of initial fadeout will be different to that as described in the Discussion.

Response 2.4

Our current framework does not consider simultaneous arrivals of imported cases.

Comment 2.5

I believe there is insufficient detail to be able to reliably reproduce the Path reduction mentioned as part of Sections 5 and 6. Similarly, I think the detail of the determination of the 25 "fastest" paths could be increased.

Response 2.5

We have revised sections 5 and 6 with more details as follows.

“*Path reduction.* Consider the path $\psi : i \rightarrow j \rightarrow k$ in the previous section. We can approximate the pdf $E_{T_{ij}^1} \left[g_1(t | w_{jk} I_j^{CEA}) \right]$ for T_{ik}^n with $f_n(t | \lambda_\psi, \alpha_\psi)$, where λ_ψ and α_ψ are obtained by minimizing the relative entropy²⁵ for $n = 1$ (the first exportation)

$$\int_0^\infty E_{T_{ij}^1} \left[g_1(t | w_{jk} I_j^{CEA}) \right] \ln \left(\frac{E_{T_{ij}^1} \left[g_1(t | w_{jk} I_j^{CEA}) \right]}{f_1(t | \lambda_\psi, \alpha_\psi)} \right) dt .$$

This is a simple 2-dimensional optimization problem. The accuracy of such path reduction (**Fig. 2f** and **Fig. S5**) implies that the spread of epidemics from the origin to any population $k \in D_{i,2}$ can be regarded as a two-population model, in which (i) the adjusted mobility rate is α_ψ and (ii) the epidemic in the origin grows exponentially at rate λ_ψ . Next, consider the path $\varphi : i \rightarrow j \rightarrow k \rightarrow m$, i.e. $m \in D_{i,3}$. Using path reduction, we can approximate φ with $\varphi' : i \rightarrow k \rightarrow m$ where the adjusted mobility rate and epidemic growth rate in the origin for the $i \rightarrow k$ leg are α_ψ and λ_ψ , respectively. The arrival times of imported cases in population $m \in D_{i,3}$ (i.e. T_{im}^n , $n = 1, 2, \dots$) can then be estimated using the tools (i.e. adjustments for hub-effect and continuous seeding) that we have developed for $D_{i,2}$ populations. The arrival times of imported cases for population $D_{i,c}$, $c = 4, 5, \dots$, can be estimated analogously.”

(Page 23, line 465)

“*Superposition of paths.* Let population i be the epidemic origin and consider population $k \in D_{i,c}$, i.e. population k is c degrees of separation from the epidemic origin²⁴. Superposition of NPPs for paths connecting population i to k is implemented as follows. As in the main text, let Ψ_{ik} be the

set of all acyclic paths connecting the epidemic origin to population k . Enumeration of all paths in Ψ_{ik} for every population in the WAN is computationally prohibitive⁴⁹ (and unnecessary).

Instead, we approximate Ψ_{ik} with the 25 “fastest” paths from population i to k that are identified using the following algorithm:

1. Use the depth-first search algorithm⁴⁹ to identify the set of acyclic paths from the epidemic origin to population k that have at most $c+2$ connections. We denote this set by Ω_{ik} and assume that all the paths not in Ω_{ik} have negligible contribution to the EAT for population k .
2. Define the distance between any two directly connected populations a and b as $-\ln(w_{ab})$, which is analogous to the distance metric in Brockmann and Helbing²⁰, namely $1 - \ln(w_{ab} / \sum_b w_{ab})$. We choose to use this distance metric because (as described in section 4) if population j is directly connected to population i , then $E[T_{ij}^1] \approx \frac{1}{\lambda_i} [\ln(\lambda_i / \alpha_{ij}) - \gamma]$ given $\alpha_{ij} \ll \lambda_i$, where γ denotes the Euler constant and $\alpha_{ij} = s_i w_{ij}$. This indicates that the expected EAT is proportional to $-\ln(w_{ij})$.
3. Based on our distance metric in step 2, identify the 100 shortest paths in Ω_{ik} by sorting in an ascending order. Denote the resulting set by Ω_{ik}^S .
4. For each path $\psi \in \Omega_{ik}^S$, use hub-effect adjustment, continuous-seeding adjustment and path reduction developed in the WAN-SPT analysis to calculate λ_ψ and α_ψ and the corresponding expected EAT, namely $\frac{1}{\lambda_\psi} \exp\left(\frac{\alpha_\psi}{\lambda_\psi}\right) E_1\left(\frac{\alpha_\psi}{\lambda_\psi}\right)$.
5. Approximate Ψ_{ik} with the 25 paths in Ω_{ik}^S that have the smallest expected EATs computed in step 4 (i.e. the 25 “fastest” paths). We choose to use the 25 fastest paths in Ω_{ik}^S to approximate Ψ_{ik} because Fig. S9 shows that the accuracy of EAT estimates would slightly worsen if we use only the 10 fastest paths in Ω_{ik}^S while there is little improvement in performance if we use the 50 fastest or all paths in Ω_{ik}^S .

(Page 24, line 483)

Figure S9. Superposition of paths in the WAN. Analogous to Fig. 3 with Hong Kong as the epidemic origin, this figure shows the effect of increasing the number of "fastest" paths for superposition on estimating the epidemic arrival times in the WAN. From top to bottom, the epidemic arrival time for each population in the WAN is computed with the superposition of the 10, 25, 50, and 100 "fastest" paths, respectively.

Comment 2.6

Further detail is required regarding the Markov Chain Monte Carlo inference. All parameters being inferred (or assumed known) should be detailed, along with precisely defining their prior distribution. The burn-in, convergence and mixing of chains should be discussed. What sample size has been used, and why? I also don't understand why the inferred basic reproductive number is monotonically increasing with time (this seems suspicious to me), and how this is actually consistent with the existing literature (Figure 4).

Response 2.6

Following your suggestion, we have substantially revised sections 8 and 9 of Methods by explicitly documenting the parameters being inferred, the priors, the burn-in, the convergence diagnostics, the number of MCMC iterations and the underlying rationale. Please see the revised manuscript for details (page 28, line 562 for the first case study, and page 32, line 639 for the second case study).

We apologize for the confusion on Figure 4a. Figure 4a does not imply that the inferred R_0 is monotonically increasing with time. The x-axis of Figure 4a refers to the (assumed) epidemic start time in Greater Mexico City which has been estimated to be between 18 February and 14 March 2009 in surveillance reports²⁹ and other studies^{27,28,30,31}. As such, Figure 4a shows that the inferred R_0 increases if the epidemic started in Greater Mexico City later.

Reviewers' comments:

Reviewer #1 (Remarks to the Author):

The authors have addressed my concerns in this revision. In particular, they now describe the work as a "framework" rather than a "theory" (which better describes the scope of the results) and make explicit the importance of each assumption in the framework early in the Discussion. I believe this version is suitable for publication.

Reviewer #2 (Remarks to the Author):

I have one remaining concern, and that is with respect to my original Comment 2.2 and the response to it.

It is probable that I am confused, but I still have issue with the removal of the rate of import to population j when considering the rate of growth in prevalence in population i .

The authors have simply added "Note that this does not mean that the disease prevalence in population i would be different dependent upon which population we are viewing it from. " to the explanation, but this isn't satisfactory to me.

Shouldn't the disease prevalence in population i be growing exponentially at rate $\lambda_i - \sum_k w_{ik}$?

Under your approximation, the exponential growth rate is different in population i as you consider each patch, j , it is connected to with a different w_{ij} .

Further, in the same explanation, the authors write "Because w_{ij} is not small compared to λ_i but large compared to w_{ik} , we expect...". The example has $w_{ij} = 0.1\lambda_i$, and $w_{ik} = 0.01\lambda_i$.

Point-by-point response (NCOMMS-17-09001A)

Reviewer 1

Comment 1.1

The authors have addressed my concerns in this revision. In particular, they now describe the work as a "framework" rather than a "theory" (which better describes the scope of the results) and make explicit the importance of each assumption in the framework early in the Discussion. I believe this version is suitable for publication.

Response 1.1

Thank you for your supportive comments.

Reviewer 2

Please note: Page and line numbers refer to those in the cleaned manuscript.

Comment 2.1

I have one remaining concern, and that is with respect to my original Comment 2.2 and the response to it.

It is probable that I am confused, but I still have issue with the removal of the rate of import to population j when considering the rate of growth in prevalence in population i .

The authors have simply added "Note that this does not mean that the disease prevalence in population i would be different dependent upon which population we are viewing it from. " to the explanation, but this isn't satisfactory to me.

Shouldn't the disease prevalence in population i be growing exponentially at rate $\lambda_i - \sum_k w_{ik}$?

Under your approximation, the exponential growth rate is different in population i as you consider each patch, j , it is connected to with a different w_{ij} .

Further, in the same explanation, the authors write "Because w_{ij} is not small compared to λ_i but large compared to w_{ik} , we expect...". The example has $w_{ij} = 0.1\lambda_i$, and $w_{ik} = 0.01\lambda_i$.

Response 2.1

We apologize for the confusion. We have revised the "Hub-effect" section in Methods Section 5 as follows to clarify this point. In particular, we have removed the example because it did not seem to be helpful.

“*Hub effect*. Suppose the epidemic origin (population i) is directly connected to one or more populations, one of which is population j (as illustrated in **Fig. 2a**). In the deterministic version of our metapopulation epidemic model (see Section 1), the disease prevalence in population i during the exponential growth phase is well approximated by the differential equation

$$\frac{dI_i}{dt} = \lambda_i I_i - \sum_k w_{ik} I_i = \left(\lambda_i - \sum_{k \neq j} w_{ik} \right) I_i - w_{ij} I_i$$

where the actual growth rate of the disease prevalence in population i is $\lambda_i - \sum_k w_{ik}$. This differential equation leads us to make the following conjecture: In our original stochastic model, in which the epidemic arrival process for population j is essentially an NPP with intensity function being the second term of the above equation (i.e. $w_{ij} I_i$), we can estimate the EAT for population j using the results from the two-population model (Section 4) in which population i is exporting cases to population j at mobility rate w_{ij} (viewed as a stochastic process) and the disease prevalence in population i is growing exponentially at rate $\lambda_{ij} = \lambda_i - \sum_{k \neq j} w_{ik}$ (viewed as a deterministic process). The hub-adjusted growth rate λ_{ij} can be interpreted as the rate at which disease prevalence in population i is growing exponentially before population j imports its first case from population i . Note that the hub-adjusted rate $\lambda_{ij} = \lambda_i - \sum_{k \neq j} w_{ik}$ is not the same as the actual growth rate, namely $\lambda_i - \sum_k w_{ik}$. To see this, consider the two-population model in which population i is only connected to population j . In this case, the EAT distribution is

given by equation (1) which requires λ_{ij} to be the hub-adjusted rate $\lambda_i - \sum_{k \neq j} w_{ik} = \lambda_i$ but

not the actual growth rate $\lambda_i - \sum_k w_{ik} = \lambda_i - w_{ij}$.”

(Page 21, line 418)

REVIEWERS' COMMENTS:

Reviewer #2 (Remarks to the Author):

The new explanation is improved. The emphasis on before the first import to j is important.